**Spatial variability of organic matter molecular composition and elemental geochemistry in surface sediments of a small boreal Swedish lake**

Julie Tolu[1*], Johan Rydberg[1], Carsten Meyer-Jacob[1], Lorenz Gerber[2] and Richard Bindler[1]

[1] *Department of Ecology and Environmental Science, Umeå University, SE-901 87 Umeå, Sweden*

[2] *Umeå Plant Science Center, Swedish University of Agricultural Sciences, Department of Forest Genetics and Plant Physiology, SE-901 83 Umeå, Sweden*

\* Corresponding author. julietolu@hotmail.com

**Abstract.**
The composition of sediment organic matter (OM) exerts a strong control on biogeochemical
processes in lakes, such as those involved in the fate of carbon, nutrients and trace metals.
While between-lake spatial variability of OM quality is increasingly investigated, we explored
in this study how the molecular composition of sediment OM varies spatially within a single
lake, and related this variability to physical parameters and elemental geochemistry. Surface
sediment samples (0-10 cm) from 42 locations in Härsvatten – a small, boreal forest lake with
a complex basin morphometry – were analyzed for OM molecular composition using
pyrolysis gas chromatography/mass spectrometry, and for the contents of twenty-three
major/trace elements and biogenic silica. 162 organic compounds belonging to different
biochemical classes of OM (e.g., carbohydrates, lignin and lipids) were identified. Close
relationships were found between the spatial patterns of sediment OM molecular composition
and elemental geochemistry. Differences in the source types of OM (i.e. terrestrial, aquatic
plant and algal OM) were linked to the individual basin morphometries and chemical status of
the lake. The variability in OM molecular composition was further driven by the degradation
status of these different source pools, which appeared to be related to sedimentary physico-
chemical parameters (e.g., redox conditions) and to the molecular structure of the organic
compounds. Given the high spatial variation in OM molecular composition within Härsvatten
and its close relationship with elemental geochemistry, the potential for large spatial
variability across lakes should be considered when studying biogeochemical processes
involved in the cycling of carbon, nutrients and trace elements or when assessing lake
budgets.
**Keywords.**
Lake sediment; spatial variability; organic matter; molecular composition; Py-GC/MS;
elemental geochemistry

## 1. Introduction

In lake basins, a wide range of factors are known to influence the transport and fate of sedimentary material, such as the location of inlet streams, catchment topography, land-use patterns, fetch, basin morphometry and sediment focusing. Sediment focusing results from a combination of factors such as wind and wave action, basin slope and the settling velocity of different particle sizes, which all contribute to the redistribution of light, fine-grained material rich in clays, organic matter (OM) and associated trace elements from shallower to deeper waters (Blais and Kalff, 1995; Ostrovsky and Yacobi, 1999). While sediment focusing is important, catchment and lake characteristics can be complex and exert a primary influence on spatial patterns in sediment geochemistry, such as in relation to land use in near-shore areas (Dunn et al., 2008; Vogel et al., 2010; Sarkar et al., 2014), complex lake/basin morphometries (Bindler et al., 2001; Rydberg et al., 2012) or river inflows (Kumke et al., 2005). The presence of macrophytes or wind-induced water currents have also been shown to affect the spatial distribution of e.g., lead (Pb), phosphorus (P) and OM (Benoy and Kalff, 1999; Bindler et al. 2001).

Because trace metals and nutrients are primarily associated with – or are part of – OM, studies focusing on the spatial patterns of metal or nutrient accumulation typically include an analysis of the OM content. The two standard approaches to determine sediment OM content are the analysis of loss-on-ignition (LOI; Ball, 1964; Santisteban et al., 2004) or the analysis of elemental carbon (C). However, either approach inherently treats OM as a homogeneous sediment component. Recent studies interested in the role of lake sediments as a long-term C sink have likewise mainly treated OM and C as a homogeneous component (e.g., Sobek et al., 2003; Tranvik et al., 2009; Heathcote et al., 2015). Even if this approach is rational from a global perspective of calculating C budgets, treating OM as a homogeneous component is overly simplistic from the perspective of developing insights into the biogeochemical behavior of OM and its influence on C, nutrients and

trace metals cycling, and does not take full advantage of the information provided by differences in
the OM quality.
In boreal lakes the sediment composition is often dominated by OM, typically ranging from 20
to 60 % on a dry weight basis. Biogenic silica (bSi) may account for as much as 45 % of the
sediment dry weight (Meyer-Jacob et al., 2014), and the remaining sediment mainly consists of
detrital mineral matter and possibly authigenic minerals. Lake OM is an extremely heterogeneous
and complex mixture of molecules that are derived from residues of plants, animals, fungi, algae
and micro-organisms, and which are either transported into the lake from the surrounding
catchment (allochthonous) or produced within the lake (autochthonous). Furthermore, these organic
compounds may undergo transformations within the water column and the sediment through both
biotic and abiotic processes. There have been a few studies where the spatial complexity in OM
quality within a lake basin has been assessed using infrared spectroscopy, which yields qualitative
information on variations in OM quality (Korsman et al., 1999; Rydberg et al., 2012), or
quantitative analyses of photopigments and lipids (Ostrovsky and Yacobi, 1999; Trolle et al., 2009;
Vogel et al., 2010; Sarkar et al., 2014). However, little work has been done to detail how the
molecular composition of the sediment OM matrix varies spatially within a lake, considering a
large number of organic biochemical classes and compounds.
To characterize OM composition at the molecular level, the most commonly used methods are
based on liquid or gas chromatography (LC or GC) coupled to fluorescence or mass spectrometry
(MS) detection. These methods provide quantitative data on original organic compounds found in
the analyzed samples, including highly specific biomarkers of, e.g., OM sources, and have been
successfully employed to study OM composition and reactivity in environmental matrices as well
as to reconstruct environmental changes (e.g., changes in vegetation, algal productivity) from peat
or  sediment  cores.  However,  the  associated  sample  preparation  procedures,  i.e.
extraction/hydrolysis and derivatization, are fastidious and specific to the different biochemical
classes of organic compounds such as carbohydrates, proteins/amino acids, lipids, chlorophylls and
lignin (e.g., Wakeham et al., 1997; Dauwe and Middelburg, 1998; Tesi et al., 2012). Moreover,
sample masses > 10 mg are required. Hence, studies where different OM biochemical classes are
targeted using these wet chemical extraction and GC/LC-MS methods are very scarce. But, efforts
in characterizing the whole OM composition at the molecular level can bring important insights
because the different biochemical classes of OM do not always include specific biomarkers for the
different existing sources of OM (e.g., terrestrial plants, macrophytes, higher plants, mosses, algae,
bacteria). For example, lignin oligomers are only specific of higher plants (Meyer and Ishiwatari,
1997) and proteins/amino-acids mainly provide biomarkers for bacteria and planktonic production
(Bianchi and Canuel, 2011). Moreover, the different biochemical classes of OM do not present the
same reactivity; for example, proteins/amino-acids and neutral carbohydrates have been shown to
be among the most reactive organic molecules (e.g., Fichez, 1991; Dauwe and Middelburg, 1998;
Amon and Fitznar, 2001; Tesi et al., 2012). Advanced ultrahigh-resolution MS techniques, i.e.
Fourier transformed-ion cyclotron resonance-mass spectrometry (FT-ICR-MS) or Linear trap
quadruple-orbitrap-MS, enable the determination of a large number of organic molecular formulas
in liquid samples (> 1000; e.g., Hawkes et al., 2016). These method have been successfully used to
link variability in the molecular composition of dissolved OM (DOM) with different factors and/or
processes of environmental ecosystems, such as climate, hydrology and OM degradation in boreal
lakes (Kellerman et al., 2014; Kellerman et al., 2015) or optical properties and DOM photo-
chemical alterations in wetland and seawater (Stubbins and Dittmar, 2015; Wagner et al., 2015).
However, in addition to the limited access to these advanced MS techniques due to instrumental
costs, extraction/hydrolysis steps are required when studying solid samples, which make these
methods also specific to the different biochemical classes of organic compounds.
To study the variability of OM composition in sediments, pyrolysis-gas chromatography/mass
spectrometry (Py-GC/MS) is a good compromise between (i) the quantitative LC/GC-MS or the
high-resolution MS methods that target specific compounds and, (ii) the qualitative, non-molecular
information provided by high-throughput techniques such as infrared spectroscopy or 'RockEval'

pyrolysis. Py-GC/MS analysis requires no complex sample preparation, but yields semi-quantitative data on >100 organic compounds that are chemical fingerprints of the different OM biochemical classes, which include specific biomarkers for OM sources and degradation status (Faix et al., 1990; Faix et al., 1991; Peulvé et al., 1996; Nierop and Buurman, 1998; Schulten and Gleixner, 1999; Lehtonen et al., 2000; Nguyen et al., 2003; Page, 2003; Buurman et al., 2005; Fabbri et al., 2005; Kaal et al., 2007;Vancampenhout et al., 2008; Schellekens et al., 2009; Carr et al., 2010; Buurman and Roscoe, 2011; De La Rosa et al., 2011 ; McClymont et al., 2011; Micić et al., 2011; Stewart, 2012).

In the present study, we apply our newly optimized Py-GC/MS method to characterize the molecular composition of natural OM in surface sediments (0-10 cm) from 42 locations within the lake basin of Härsvatten. Härsvatten is a small boreal forest lake in southwestern Sweden that was previously studied for the spatial distribution of Pb and OM contents (Bindler et al., 2001). Our objective here was to comprehensively investigate how the molecular composition of sediment OM varies spatially across a lake with several basins. Our specific research questions were: (i) what are the spatial patterns within a single lake for various organic biochemical classes and compounds?; (ii) how does the spatial pattern of the OM molecular composition relate to physical parameters (i.e., bulk density and water depth) and elemental, inorganic geochemistry of the sediment material?; (iii) which factors or processes (e.g., provenance, transport pathway and mineralization) appear to explain the in-lake spatial variability of the OM molecular composition?

## 2. Materials and Methods

### 2.1 Study site and samples

Härsvatten is a boreal forest lake located in southwestern Sweden (58°02' N 12°03' E) in the Svartedalen nature reserve. This culturally acidified, clear-water, oligotrophic and fishless lake has been intensively monitored since the 1980's (national database, Dept. of aquatic sciences and assessment, Swedish university of agricultural sciences, Uppsala, Sweden; www.slu.se), during which time the pH has ranged from 4.2-4.5 in 1983–1987 to 4.7-5.6 in 2010–2014. The lake is dimictic with a thermal stratification between 10 and 15 m depth in the summer. Approximately 80 % of the lake bottom is within the epilimnion. The surface areas of the lake and its catchment are 0.186 and 2.03 km$^2$, respectively. The catchment is characterized by an uninhabited, coniferous-dominated forest (*Picea abies* Karst. and *Pinus sylvestris* L.), which extends to the rocky shoreline. The bedrock consists of slow-weathering granites and gneisses that are covered by thin and poorly developed podsolic soils.

The basin of Härsvatten can be divided into four general areas (Bindler et al., 2001): 1) the main south basin, which represents about half of the lake area (sample sites S1–24; maximum depth, 24.3 m) and includes the lake's small outlet stream; 2) a north basin (sample sites N1–11; maximum depth, 12 m), which includes a small inlet stream draining from the headwater lake Måkevatten that enters Härsvatten through a small wetland; 3) an east basin, which has a maximum depth of nearly 10 m (sample sites E1–6) and is separated from the main north–south axis of the lake by a series of islands and shallow sills (<3 m water depth); and 4) a generally shallow (<3 m water depth) central area separating the north/east and south basins (sample sites M1–6).

In total, we analyzed 44 surface sediment (0–10 cm) samples that were collected in winter 1997–1998 (Fig. 1) for a study of Pb and spheroidal carbonaceous particles (Bindler et al., 2001). These samples were collected as follow: short sediment cores (0-25 cm) were taken with a gravity

corer from the ice-covered lake in winter 1997 and 1998, and were sectioned on-site into an upper
sample (0-10 cm) and a lower sample (10-25 cm; not studied here). In the laboratory, the samples
were weighed, freeze-dried, and reweighed to determine the water content and dry mass of the
sediment. The freeze-dried samples have been stored in plastic containers within closed boxes
shielded from light and at room temperature since winter 1997-1998. Before further analysis in this
study, the samples were finely ground at 30 Hz for 3 min using a stainless steel "Retsch" swing
mill.

2.2 Major and trace elements concentrations

The concentrations of major (Na, Mg, Al, Si, K, Ca, P, S, Mn, Fe) and trace elements (Sc, Ti,

V, As, Br, Y, Zr, Ni, Cu, Zn, Sr, Pb) were determined using a wavelength dispersive X-ray
fluorescence spectrometer (WD-XRF; Bruker S8 Tiger) and a measurement method developed for
powdered sediment samples (Rydberg, 2014). Accuracy was assessed using sample replicates,
which were within ±10 % for all elements.

Total mercury (Hg) concentrations were determined using thermal desorption atomic

absorption spectrometry (Milestone DMA80) with the calibration curves based on analyses of
different masses of four certified reference materials (CRMs). Analytical quality was controlled
using an additional CRM and replicate samples included with about every ten samples. The CRM
was within the certified range, and replicate samples were within ±10 % for Hg concentrations <30
$\mu$g kg$^{-1}$ and within ±5 % for concentrations ≥30 $\mu$g kg$^{-1}$.

We also included the OM content (in % dry mass), determined as loss-on-ignition (LOI) after

heating dried samples at 550ºC for 4 h in the earlier study of Bindler et al. (2001).
2.3 Biogenic silica concentrations

Biogenic silica (bSi) was determined by Fourier transform infrared (FTIR) spectroscopy
following the approach described in Meyer-Jacob et al. (2014). In brief, sediment samples were
mixed with potassium bromide (0.011 g sample and 0.5 g KBr) prior to analysis with a Bruker
Vertex 70 equipped with a HTS-XT accessory unit (multisampler). The recorded FTIR spectral
information were used to determine the bSi concentrations employing a PLSR calibration based on
analyses of synthetic sediment mixtures with defined bSi content ranging from 0 to 100 %.
We calculated the mineral Si fraction ($Si_{mineral}$) from the difference between the total Si
concentration determined by WD-XRF (Sect. 2.2) and the bSi concentration.

2.4 Organic matter molecular composition

The molecular composition of OM was determined by pyrolysis-gas chromatography-mass
spectrometry (Py-GC/MS) following the method developed by Tolu et al. (2015). In brief, 200 ±
10 µg sediment was pyrolyzed in a FrontierLabs PY-2020iD oven (450 °C) connected to an Agilent
7890A-5975C GC/MS system. Peak integration was done using a data processing pipeline under
the 'R' computational environment. Peak identification was made using the software 'NIST MS
Search 2' containing the library 'NIST/EPA/NIH 2011' and additional spectra from published
studies.
In the sediments of Härsvatten, 162 Py-products were identified, and peak areas were
normalized by setting the total identified peak area for each sample to 100 %. A detailed list of the
162 identified organic compounds is provided in the supplementary information along with
information on their molecular mass and structure, references for the theoretical mass spectra and
calculated or reference retention index values (Table S1). Although the pyrolysis temperature we
employed, i.e. 450°C as used in plant science (e.g., Faix et al., 1990; Faix et al., 1991), is different
from the pyrolysis temperature which was most commonly used for analyzing soils, sediments, peat
records and algae (i.e. >600°C), our list is highly similar to published lists of identified pyrolytic
organic compounds both in terms of the organic compounds and of their classification into 13 OM
classes (Faix et al., 1990; Faix et al., 1991; Peulvé et al., 1996; Nierop and Buurman, 1998;
Schulten and Gleixner, 1999; Lehtonen et al., 2000; Nguyen et al., 2003; Page, 2003; Buurman et
al., 2005; Fabbri et al., 2005; Kaal et al., 2007;Vancampenhout et al., 2008; Schellekens et al., 2009
;Carr et al., 2010; Buurman and Roscoe, 2011; De La Rosa et al., 2011 ; McClymont et al., 2011;
Micić et al., 2011; Stewart, 2012). Pyrolysis at 450 °C is preferred to pyrolysis at 650 °C when
using very small sample mass (few hundred μg) because it avoids complete degradation of some
specific biomarkers of OM sources and enables determination of OM degradation status by
identification of levosugars (Py products of polysaccharides and/or cellulose) or syringol lignin
oligomers, for instance (Tolu et al., 2015).

2.5 Statistical analysis

We performed all statistical analyses using SPSS software package PASW, version 22.0. Two

separate principal component analyses (PCA) were performed, one for the elemental geochemistry
(i.e., dry bulk density (B.D.) and contents of OM (LOI), major/trace elements and bSi) and the
other for the OM molecular composition. Prior to the PCA, all data were converted to Z-scores
(average = 0, variance = 1). Principal components (PCs) with eigenvalues > 1 were retained and
extracted using a Varimax rotated solution. Factor loadings were calculated as regression
coefficients, which is analogous to $r$ in Pearson correlations. For convenience the loadings are
reported as percentage of variance explained, i.e., as squared loadings. For all PCs, variables with
squared loadings <0.15 are not discussed with respect to that PC. Others variables, e.g., water depth
(W.D.) or ratios between elements, were included passively in the PC-loadings plots by using bi-
variate correlation coefficients between these variables and the PC-scores of each PC. Hierarchical

agglomerative cluster analysis (CA) was performed for the elemental geochemistry and the OM

molecular composition datasets using Wards linkages (Ward, 1963) based on squared Euclidean

distances. The PC-scores from the PCAs were used instead of the original data in order to eliminate

the effects of autocorrelation in the dataset.

**3. Results and discussion**

3.1. Sediment elemental geochemistry

*3.1.1 General description and trends*

Summary statistics of the elemental geochemical properties of the surface sediments from

Härsvatten are presented in Table 1 and the detailed data are given in Table S2 in the SI. The

sediments from sites M4 and S15 are two outliers because they have a B.D., bSi, OM and elemental

contents (e.g., Na, Mg, Al, K) that deviated by more than four standard deviations from the average

values of all analyzed sediment samples (Table 1). Moreover, these sediment samples are too coarse

(predominantly sand) for Py-GC/MS analysis according to our method based on $200 \pm 20$ µg

analyzed sample mass. Hence, they are excluded from the statistical analyses and discussion. Even

when excluding these two sites, the elemental geochemical parameters vary considerably across the

lake basin, with Hg, Fe, Co and Mn contents illustrating the greatest variabilities (i.e., coefficient of

variation, CVs >60 %) and Al, Br, K, Ti, V, Ni, Mg and Ca contents showing the lowest

variabilities (CVs: 17-25 %; Table 1). For most of geochemical properties, the average to median

ratios are approximatively 1.0, indicating no extreme values. Slightly higher values were, however,

observed for P, Fe, As and Co contents (1.2-1.3), and Mn content is associated with extremely large

values outside the population distribution (average:median = 4.1).

The lowest B.D. is observed among the three deepest sampling locations (23.5-24.5 m) in the
main south basin, where we also find the lowest bSi content and the highest contents in organically
bound elements including S, Br, P and certain trace metals, i.e. Cu, Ni, Hg and Zn. These sediments
have high OM content (> 50 %), but the highest [OM] (57-58 %) are observed among isolated sites
that are located close to the shoreline (N1-2, E3, S5, S23; 3.1-7.4) and which also include the
lowest [Al], [P], [K], [$Si_{inorganic}$], [V] and [Zr]. The highest B.D. and the lowest [OM], [S], [Br],
[Cu], [Ni], [Hg] and [Zn] are observed among the shallow sites (1.8-2.5 m) located between the
north and east basins and between the larger north and south basins (i.e., sites N10, M1, M5-6),
which also contain the highest [bSi], [Sr], [Al], [Y], [Mn] and [Co]. The sediments located at
intermediate water depth (9-20 m) in the main south basin (S4, S9, S11, S13-14, S17, S19, S22) are
associated with the highest [Fe], [As], [K], [Mg], [Na], [Ti] and [Zr], while among the shallower
sites of the south basin we find the highest [$Si_{inorganic}$]. The lowest [Fe], [As], [Co] and [Y] are
observed among the sediments of the east basin, and the sediments of the north basin include the
lowest [Mn], [Ca], [K], [Mg], [Na], [Sr] and [Zr]. To identify more accurately the most significant
relationships between the different elemental geochemical properties and to explore more precisely
their spatial distribution, the results of PCA and cluster analyses are further presented and
discussed.

*3.1.2. Principal components of the elemental geochemistry*

For the elemental geochemistry dataset, five principal components were retained. We present
only the first four PCs, which together explain 74 % of the total variance (PC1-4$_{geo}$; Fig. 2), because
no reasonable interpretation could be made for PC5$_{geo}$ (10 % of the total variance; Fig. S1 in the
SI). PC1$_{geo}$ captures 25 % of the total variance and separates bSi and B.D. (negative loadings) from
OM, S, Cu, Hg, Ni, Zn and, to a lesser extent, As and Pb (positive loadings; Fig.2a). This means
that bSi and B.D. are significantly positively correlated, and both are significantly negatively

correlated to OM, S, Cu, Hg, Ni, Zn and, to a lesser extent, As and Pb. If those parameters do not have significant loadings on PC2-5, it means they are not significantly correlated with the parameters found on PC2-5, the PCs being orthogonal to each other. The negative loadings on PC1 are interpreted as reflecting a bSi-rich fraction, while positive loadings indicate an organic-rich fraction that is enriched in organophilic trace metals (Lidman et al., 2014). For $PC2_{geo}$, which captures 21 % of the total variance, $Si_{inorganic}$, K, Na, Mg, Zr and Ti have positive loadings, while no element is significantly negatively correlated to $PC2_{geo}$ (Fig. 2a). High $PC2_{geo}$ scores likely represent samples that are richer in silicate minerals such as quartz and clays (Koinig et al., 2003; Taboada et al., 2006).

Positive loadings on $PC3_{geo}$, which explains 16 % of the total variance, are found for Al and Fe along with As, P and Y (Fig. 2b). Compared to elements such as Mg, Na and K that are mostly confined to the silicate fraction of sediments, Fe and Al may reflect both detrital material and dissolved or amorphous phases. However, the fact that As and P contents as well as the Fe:Al ratio plot together with Fe and Al contents on the positive side of $PC3_{geo}$ and not with the S content strongly suggest that sediments with high $PC3_{geo}$ scores are associated with higher contents of Fe and Al (oxy)hydroxides, which are known to strongly bind both As and P (Mucci et al., 2000; Plant et al., 2005; Zhu et al., 2013). $PC4_{geo}$ captures 12 % of the total variance and separates Mn, Co, Pb and to a lesser extent Fe (positive loadings) from OM and Br (negative loadings; Fig. 2b). Although Mn, like Fe and Al, is not confined to a specific mineral phase and can reflect both detrital or dissolved and amorphous phases, the positive loadings are interpreted as reflecting Mn (oxy)hydroxides, which bind Pb, especially when they contain cobalt (Co) (Yin et al., 2011). This interpretation is supported by the positive loadings on $PC4_{geo}$ of the ratio Mn:Fe, often used as a paleolimnological proxy for bottom water oxygenation (Naeher et al., 2013). The negative loadings could indicate a terrestrial OM fraction that is rich in Br (Leri and Myneni, 2012).

 *3.1.3 Cluster analysis of the elemental geochemistry*


For the cluster analysis of the elemental geochemistry dataset, we selected a solution of six
clusters (cluster$_{geo}$ 1–6; Fig. 1c). The cluster averages and standard deviations of each physical and
geochemical variable are given in Table S3 in the SI where they are compared to the averages
values of all analyzed sediment samples, hereafter referred as 'whole-lake average'. Table 3
provides the cluster averages for a selection of geochemical parameters.
In the south basin, the sediments found at shallower water depth (cluster$_{geo}$ 6; n=10) have a
higher B.D., are richer in bSi (negative scores on PC1$_{geo}$; Fig. 2a) and have lower than whole-lake
average trace metal concentrations (Table 1). In contrast, the sediments from the deeper sites
(cluster$_{geo}$ 5; n=3) have the lowest B.D. and lowest bSi content (Table 1), and are enriched in OM
and trace metals (positive scores on PC1$_{geo}$; Fig. 2a). The sediments found at intermediate water
depths (cluster$_{geo}$ 2; n=8) have positive scores on PC2$_{geo}$ (Fig. 2a), and they have an OM content
within 10 % of whole-lake average while trace metal concentrations are above 10 % of whole-lake
averages (Table 1). The south basin as a whole has higher P concentrations than the north, east and
central areas, and in both intermediate and deeper sites, the sediments are rich in Fe and As
(positive scores on PC3$_{geo}$; Fig. 2b and Table 1).
The sediments found at shallow water depth between the north and east basins and in the
central area (cluster$_{geo}$ 3; n=4) have the highest B.D. and are the most enriched in both bSi (negative
score on PC1$_{geo}$; Fig. 2a) and Mn and Fe (oxy)hydroxides (positive score on PC4$_{geo}$; Fig. 2b). A
small number of shallow, near-shore sampling locations (cluster$_{geo}$ 4; n=4) have higher OM
concentrations than the whole-lake average, and are enriched in S and trace metals (positive scores
on PC1$_{geo}$; Fig. 2a and Table 1).
3.2 Sediment organic matter molecular composition

*3.2.1 General description and trends*

The Py-products identified in the surface sediments of Härsvatten were classified into 13 OM
classes, i.e., carbohydrates, N-compounds, chitin-derived Py-products, phenols, lignin,
chlorophylls, *n*-alkenes, *n*-alkanes, alkan-2-ones, steroids, tocopherols, hopanoids, and
(poly)aromatics, in agreement with previous studies using Py-GC/MS for different environmental
matrices (Faix et al., 1990; Faix et al., 1991; Peulvé et al., 1996; Nierop and Buurman, 1998;
Schulten and Gleixner, 1999; Lehtonen et al., 2000; Nguyen et al., 2003; Page, 2003; Buurman et
al., 2005; Fabbri et al., 2005; Kaal et al., 2007;Vancampenhout et al., 2008; Schellekens et al., 2009
;Carr et al., 2010; Buurman and Roscoe, 2011; De La Rosa et al., 2011 ; McClymont et al., 2011;
Micić et al., 2011; Stewart, 2012). For the sake of making the presentation of the data and the
associated discussion more constrained and avoid over-interpreting individual compounds, the 162
identified organic compounds were reduced to 41 groups of compounds as described in Table 2.
This grouping is based on similarities in the molecular structure within the OM classes (cf. Table
S1 in the SI), and preliminary principal components analyses have shown that the compounds
within each of our 41 groups are highly positively correlated and thus present the same trends in our
study (data not shown). As an example, the 20 identified carbohydrate compounds, previously
demonstrated to derive from pyrolysis of polysaccharides and carbohydrates (Faix et al., 1991),
have been separated into 6 groups based on the number of C in the heterocycles of these
compounds and on their side-chain functional groups. The heterocycle of "furan" and "furanone"
compounds contains 4 C and 1 oxygen (O) atoms, and the side-chain is either aliphatic
((alkyl)furans and (alkyl)furanones) or contains an oxygenated functional group (hydroxy- or
carboxy-furans and furanones). While the heterocycle of "pyran" compounds has 5 C and 1 O
atoms, the one of dianhydrohamnose, levoglucosenone and levosugars consists in 6 C and 1 O. But,
the levosugars contain three hydroxyl functional groups whereas dianhydrorhamnose contains 2
hydroxyl groups and levoglucosenone have a carbonyl group instead.

Summary statistics of these 41 groups of organic compounds are presented in Table 2 and the

detailed data are given in Table S4 in the SI. The coefficients of variation for the abundances of the
different organic compound groups range from 15 to 106 % with an average of $38 \pm 20$ %, showing
a remarkable in-lake variability of OM molecular composition. For most of the organic compound
groups, the average to median ratios are approximatively 1.0, indicating no extreme values.
However, slightly higher values (1.2-1.8) are observed for organic compounds derived from higher
plants and mosses, i.e. levosugars, lignin oligomers (syringols and guaiacols), $n$-alkanes C25-35,
alkan-2-ones C23-31 and tocopherols.

Most of the N-compounds, which usually derive more from algae than from higher plants and

mosses (Bianchi and Canuel, 2011), have the highest abundances among the three deepest sampling
locations (23.5-24.5 m) in the main south basin (S12, S18 and S24). These three deepest sampling
locations also present the highest abundances of (i) pyrolytic compounds containing an acetamide
functional group previously shown to be a good indicator of the presence of chitin, a component of
fundi cell walls and arthropod exoskeletons, in biological and geological samples (Gupta et al.,
2007); (ii) phytadienes, i.e., pyrolytic products of chlorophylls (Nguyen et al., 2003); (iii) short-
chain alkan-2-ones (2K C13-17); and (iv) steroids. In contrast, most of the carbohydrates, which
usually derive mostly from higher plants and mosses (Bianchi and Canuel, 2011), have the highest
abundances among the sediments situated close to the shoreline (N1-2, E3, S5, S23) such as for the
abundances of phenols, guaiacyl- and syringyl-lignin oligomers, long-chain $n$-alkenes (C27-28:1)
and diketodipyrrole (N-compounds), all specific of higher plants and/or mosses OM (Meyers and
Ishiwatari, 1993; Schellekens et al., 2009). The highest abundances of long-chain $n$-alkanes (C23-
26:0 and C27-35:0) and mid-chain $n$-alkanes (C17-22:0) are, however, observed for the shallower
sites (<2 m) situated between the larger north and south basins (sites M5-6).
Among the shallow sites (2.5-3.0 m) located between the north and east basin (N10, M1) and
the shallow and intermediate water depth (4-20 m) sites of the south basin (S1-4, S6-11, S13-17,
S19-22), we find the highest abundances of degradation products of carbohydrates (i.e.,
(alkyl)furans & furanones and hydroxyl- or carboxy-furans & furanones), of proteins, amino-acids
and/or   chlorophylls   (i.e.,   pyridines_O,   (alkykl)pyrroles,   pyrroles_O,   pyrroledione   &
pyrrolidinedione, pristenes) and of lipids (i.e., short-chain $n$-alkenes and $n$-alkanes – C9-16:1 and
C13-16:0) as well as the highest abundances of (poly)aromatic compounds indicative of highly
degraded OM (Schellekens et al., 2009; Buurman and Roscoe, 2011). The lowest abundances of the
(poly)aromatic and certain aliphatic compounds (i.e. $n$-alkenes C17-22 and C27-28, $n$-alkanes C13-
16 and alkan-2-ones C13-17) are observed among the sediments located close to the shoreline (N1-
2, E3, S5, S23), while the two shallow sites situated between the larger north and south basins (M5-
6) present the lowest abundances for all other organic compounds. To identify more accurately the
most significant relationships between the different organic compounds groups and to explore more
precisely their spatial distribution, the results of PCA and cluster analyses are further presented and
discussed.

*3.2.2 Principal components of OM molecular composition*

For the OM molecular composition dataset, six principal components (PC1-6$_{OM}$) were retained,
which explain 85 % of the total variance (Fig. 3). PC1$_{OM}$, which captures 30 % of the total variance,
separates organic compounds that are produced during OM degradation (positive loadings), from
molecules of higher plant or moss origin including those that are readily mineralized (negative
loadings; Fig. 3a). Compounds with positive loadings include i) (poly)aromatics (i.e., benzene,
acetylbenzene, benzaldehyde, alkylbenzenes C2-9 and polyaromatics); and ii) degradation products
of carbohydrates ((alkyl)furans & furanones; Schellekens et al., 2009), proteins, amino acids and/or
chlorophylls (aromatic N, (alkyl)pyridines and (alkyl)pyrroles; Jokic et al., 2004; Sinninghe Damsté
et al., 1992) and lipids (short-chain $n$-alkanes – C13-16:0 –, $n$-alkenes – C9-16:1 – and alkan-2-
ones – 2K C13-17 –; Schellekens et al., 2009). Therefore, positive $PC1_{OM}$ scores represent samples
rich in degraded OM. The molecules of plant origin with negative $PC1_{OM}$ loadings are syringol and
guaiacol lignin oligomers that are specific for vascular plants, long-chain $n$-alkenes (C23-26:1 and
C27-28:1) deriving from lipids of higher plants and/or mosses (Meyers and Ishiwatari, 1993) and
long-chain alkan-2-ones (2K C23-31). Although alkan-2-ones C23-31 may arise with degradative
oxidation of $n$-alkanes/$n$-alkenes (Zheng et al., 2011), they are also good biomarkers for mosses
such as *Sphagnum* (2K C23-25) and for aquatic higher plants (2K C27-31) (Baas et al., 2000;
Hernandez et al., 2001; Nichols and Huang, 2007). Furthermore, negative loadings on $PC1_{OM}$ are
found for the anhydrosugars, that are Py-products of fresh, high-molecular weight carbohydrates
and cellulose from higher plants and mosses (never reported in Py-chromatograms of algae or
arthropods; Marbot, 1997; Nguyen et al., 2003; Valdes et al., 2013) as well as for the ratio
anhydrosugars: (alkyl)furans & furanones, a proxy for plant OM freshness (Schellekens et al.,
2009), have also (Fig. 3a). Thus, negative $PC1_{OM}$-loadings likely reflect a fresh pool of OM coming
from in-lake vegetation.

$PC2_{OM}$ captures 14 % of the total variance and positive loadings are associated with (i) mid-

chain $n$-alkanes/$n$-alkenes doublets that are known to be released during pyrolysis of resistant
biomacromolecules such as cutin, suberin and algaenan (Buurman and Roscoe, 2011); (ii) pristenes,
which are resistant degradation products of chlorophylls (Nguyen et al., 2003); and (iii) hopanoids,
which are high-molecular weight pentacyclic compounds of prokaryotes, especially bacteria, origin
(Meredith et al., 2008; Sessions et al., 2013). No compounds are significantly negatively correlated
to $PC2_{OM}$ (Fig. 3a). High $PC2_{OM}$ scores thus represent samples rich in organic molecules that are
resistant to degradation.

$PC3_{OM}$ explains 13 % of the total variance and separates carbohydrates and N-compounds that

are pyrolytic or degradation products of proteins, amino acids and/or chlorophylls (i.e., pyridines,
pyroledione and pyrrolidinedione) and of chitin on positive side, from aliphatic long-chain $n$-
alkanes (C23-26:0 and C27-35:0) coming from lipids of higher plants or mosses on negative side
(Fig. 3b).
On $PC4_{OM}$, which explains 13 % of the total variance, positive loadings are found for the
diketopiperazines, i.e. specific Py products of proteins or amino acids, the alkylamides and the
chlorophyll-derived phytadienes, which altogether indicate fresh algal organic residues (Peulvé et
al., 1996; Nguyen et al., 2003; Fabbri et al., 2005; Micić et al., 2010). Py products of chitin (Gupta
et al., 2007) and hopanoids, which derived from prokaryotes and mainly bacteria (Meredith et al.,
2008; Sessions et al., 2013), also have positive loadings on $PC4_{OM}$, while no compounds are
significantly negatively correlated to $PC4_{OM}$ (Fig. 3b). Therefore, $PC4_{OM}$ reflects OM input from in-
lake algae and micro-organisms (e.g., zooplankton, bacteria). Steroids, which have not yet been
reported by Py-GC/MS in aquatic matrices, have positive loadings on this $PC4_{OM}$ suggesting that
the steroids released by pyrolysis in aquatic samples are mainly of algal origin.
For $PC5_{OM}$, capturing 8 % of the total variance, positive loadings are related to lignin
oligomers, which are specific for vascular plants (Meyers and Ishiwatari, 1993), and
diketodipyrrole, a N-compound often reported in soil pyrolysates (e.g., Schellekens et al., 2009;
Buurman and Roscoe, 2011). No compounds are associated with negative loadings on $PC5_{OM}$ (Fig.
3c). Interestingly, the long-chain *n*-alkanes from higher plants or mosses lipids do not have positive
loadings on $PC5_{OM}$. We therefore interpret $PC5_{OM}$ to relate to OM inputs from the forested
catchment, which is dominated by coniferous species. Coniferous trees generally have higher lignin
contents as compared to other vascular plants (Campbell and Sederof, 1996), while they contain
much lower amounts of *n*-alkanes than other plant species (Bush and McInerney, 2013).
$PC6_{OM}$ captures 7 % of the total variance and has four compounds with significant positive
loadings, i.e., benzene, two benzenes with oxidized side-chain and carboxy- or hydroxy-furans and
furanones, i.e. furan and furanone heterocycles with an O atom in the side-chain (Fig. 3c). $PC6_{OM}$
may thus represent an intermediate degradation status of higher plants and/or mosses residues,
between the lignin oligomers or anhydrosugars (fresh) and the degraded polyaromatics and
benzenes C2-9 or (alkyl)furans and furanones (i.e. furan and furanone heterocycles with an
aliphatic side-chain).

*3.2.3 Cluster analysis of OM composition*

As with the elemental geochemistry dataset, a solution of six clusters (cluster$_{OM}$ 1–6) was
relevant to represent the data on the spatial heterogeneity of OM molecular composition (Fig. 1d).
Each cluster is associated with one or a few of the OM types that were identified by the PC1-6$_{OM}$
(Fig. 3; Sect. 3.2.1). The cluster averages and standard deviations of each organic compound are
given and compared to whole-lake averages in Table S5 in the SI. Table 3 provides the cluster
averages for ratios indicative of OM source types and their degradation status based on literature
data and on the distribution of the organic compounds on PC1-6$_{OM}$.
In the south basin, the majority of sites found at shallower and intermediate water depths group
in cluster$_{OM}$ 3 (n=14) and are enriched in degraded and resistant OM (positive scores on PC1$_{OM}$;
Fig. 3a). The deep basin sites (cluster$_{OM}$ 2; n=3) are enriched in fresh algal and zooplanktonic OM
(positive scores on PC4$_{OM}$; Fig. 3b). Accordingly, the values for the ratios indicative of higher
proportions of fresh, labile algal OM, based on N-compound or chlorophyll composition, are higher
in the deeper sites as compared to whole-lake averages, while the values are below or within ±10 %
of whole-lake averages in the sediments found at shallower and intermediate water depths (Table
1). In contrast, the ratios indicative of higher plant and moss OM freshness based on carbohydrate
or lignin composition have similar values, and lower as compared to whole-lake averages, for all
sediments of the south basin. Furthermore, the clusters$_{OM}$ 2 and 3 are characterized by higher values
for the ratios specific of algal versus higher plant and moss OM based on the proportions of N-
compounds versus carbohydrates or chlorophylls versus lignin and long-chain *n*-alkanes and alkan-
2-ones (Table 1). The rest of the south basin sites, fall within cluster$_{OM}$ 1 (n=1), 5 (n=2) or 4 (n=1),
which are described below.
The majority of sites in the northern half of the lake group within $cluster_{OM}$ 1 (n=15) with
isolated shallower sites falling within $clusters_{OM}$ 3 (n=1), 4 (n=2) and 5 (n=2). The sediments of
$cluster_{OM}$ 1 are rich in fresh plant (higher plants or mosses) OM coming from in-lake productivity
(negative scores on $PC1_{OM}$; Fig. 3a) and have higher values than whole-lake averages for the ratios
specific of in-lake vs terrestrial plant OM and of higher plant OM freshness (Table 1). In contrast,
the values for these ratios are below 10 % of whole-lake averages for the south basin sites,
indicating that terrestrial input is the main source of plant OM to the sediments of the main basin of
Härsvatten.
The $cluster_{OM}$ 5 represents some near-shore locations (n=4), which are enriched in OM derived
from the coniferous-forested catchment (positive scores on $PC5_{OM}$; Fig. 3c). The $cluster_{OM}$ 4 (n=4),
which groups shallow sites located close to the lake outlet (south basin, S16) and between the north
and east basins (N10 and M1), is characterized by high proportions of degraded and resistant OM
(positive scores on $PC5_{OM}$; Fig. 3a). Two shallow sites of the central area ($cluster_{OM}$ 6; n=2) show
an enrichment in aliphatic molecules derived from higher plant and moss lipids (negative loadings
on $PC3_{OM}$; Fig. 3b). Both $clusters_{OM}$ 4 and 6 have values for the ratio indicative of in-lake:terrestrial
plant OM above 10 % of the whole-lake average, while the values for the ratios specific of algal vs
higher plant and moss OM and of OM freshness based on N-compounds and carbohydrates
composition are below 10 % of whole-lake averages (Table 1). $Cluster_{OM}$ 6 differs from $cluster_{OM}$ 4
by its higher values for the ratios specific of OM freshness based on chlorophyll and lignin
composition.
3.3 Factors and processes involved in the spatial distribution of OM molecular composition

The surface sediments used in this study comprise the uppermost 10 cm. Given the inherent
variation in sedimentation rates across a lake basin, each bulk sample represents material deposited
over different timescales. We know from the developmental work for our Py-GC/MS method using
annually laminated sediments that there are transformations in OM composition within the
uppermost few cm, i.e., the first few years following deposition (Tolu et al. 2015). Thus, these bulk
sediment samples provide initial insights into the spatial variability in molecular OM composition
within a lake basin resulting from longer-term sedimentation processes (including those within the
sediment) reflecting years to decades.
The distribution of both clusters$_{geo}$ and clusters$_{OM}$ within Härsvatten shows a similar general
pattern (Fig. 1c and 1d) where a main feature is the separation of most of the sediments located in
the north and east basins (cluster$_{geo}$ 1 and cluster$_{OM}$ 1) from those found in the main south basin
(clusters$_{geo}$ 2, 5, 6 and clusters$_{OM}$ 2, 3). The other similarities are i) a separation of the sediments
within the main, south basin according to water depth, with cluster$_{geo}$ 5 and cluster$_{OM}$ 2 grouping
the deeper sites and clusters$_{geo}$ 2, 6 and cluster$_{OM}$ 3 grouping the shallow and intermediate depth
sites; and ii) a separation of the shallower sites that are located close to the shore (cluster$_{geo}$ 4 and
cluster$_{OM}$ 5) from the ones found between the north and east basins and between the central area
and the south basins (cluster$_{geo}$ 3 and clusters$_{OM}$ 4 and 6).

*3.3.1 Spatial variability in the main south basin*

As shown previously for OM (as % LOI) and Pb (Bindler et al. 2001), there is a physical and
inorganic geochemical gradient from shallower to deeper waters reflecting sediment focusing in the
south basin of Härsvatten. B.D. and bSi decrease from shallower (cluster$_{geo}$6) to intermediate
(cluster$_{geo}$2) and to deeper areas (cluster$_{geo}$5), whereas there is a progressive enrichment in organic

matter and trace elements with increasing water depth (Fig. 1c; Table 1). For example, B.D. decreases from ~0.07 to 0.03 g cm$^{-3}$ while OM and Hg increase from ~34 to 52 % and from ~230 to 920 ng g$^{-1}$, respectively, in shallower versus the deepest locations. At intermediate depths (cluster$_{geo}$2), OM, B.D., bSi and most trace metals (i.e., Cu, Ni, Hg, Zn) are between those of shallow and deep locations. Sediment focusing is thus an important process for sediment geochemistry in the large, deep basin of Härsvatten, which presents a relatively simple morphometry. According to the model of sediment focusing, the sediments found at shallower (<11 m; cluster$_{geo}$ 6), intermediate (11-21 m; cluster$_{geo}$ 4) and deeper water depths (>23 m; cluster$_{geo}$ 5) would correspond to zones of erosion, transportation and accumulation, respectively (Håkanson, 1977). The bSi decline, from ~15 to 4 %, indicates a decrease of diatom production with depth due to increasing light attenuation, and thus suggests that the diatom assemblage is dominated by benthic diatoms as shown for many acidified lakes, such as the surrounding lakes in the Svartedalen nature reserve (e.g., Andersson, 1985; Anderson and Renberg, 1992).

In this main basin of Härsvatten, OM originates from a combination of autochthonous algal production and allochthonous input (Sect. 3.2.2). The dominance of benthic diatoms in acidified lakes and the declining bSi content with depth would indicate that the algal material in deeper areas of the basin should mainly derive from resuspended benthic algal production. However, this benthic algal production is not reflected in the OM molecular composition. The sediments from shallow and intermediate water depths (cluster$_{OM}$ 3) are mainly composed of degraded and resistant OM, while the sediments from deeper sites (cluster$_{OM}$ 2) are enriched in fresh algal and zooplanktonic OM (Fig. 1d; Sect. 3.2.2). Although our results are based on the top 10 cm of sediment and thus account for different sediment ages, we suggest that the higher proportions of decomposed algal material, based on N-compound and chlorophyll composition (Table 1), at shallower and intermediate water depths (<21m) than at the deepest sites (23.5–24.5 m) reflect higher mineralization rates of OM in shallow/intermediate areas. Higher OM mineralization rates in shallow/intermediate areas are most probably due to more oxic conditions, which are known to prevail in epilimnitic and metalimnitic

sediments (Ostovsky and Yacobi, 1999); the epilimnion in Härsvatten has been assessed to extend to 10–15 m water depth. Higher OM preservation in the deeper area may also be favored by higher accumulation rates as compared to shallow/intermediate areas (as consequence of sediment focusing), but the sedimentation rates in the deeper areas of Härsvatten are nonetheless very low, with the uppermost 30 cm being deposited during the last c. 500 year (Bindler et al., 2001). Moreover, the elemental geochemistry indicates that the sites found at intermediate water depths (cluster$_{geo}$ 6; 11–21 m) correspond in the sediment focusing model to transportation zones, which experience recurrent resuspension events that favor gas exchanges and mineralization of OM (Ståhlberg et al., 2006). Occurrence of oxic conditions at intermediate depths in the south basin is supported by the higher concentrations of Fe, Mn, As, Co and P and the high Fe:Al values, this combination of parameters being often indicative of Fe and Mn (oxy)hydroxides (Table 1; Sect. 3.1.1). In line with our hypothesis, higher OM mineralization rates in oxic versus anoxic sediments have previously been reported (Bastviken et al., 2004; Isidorova et al., 2016). However, in contrast to the more algal-derived OM, we do not observe significant differences between the sediments of shallower/intermediate water depths and the deepest sites for ratios indicative of higher plant and moss OM freshness (Table 1). Because higher plant and moss OM is mainly of allochthonous origin in this basin, our results indicate that primarily autochthonous algal OM is mineralized in the epilimnitic and metalimnitic sediments of this deeper, steeper-sloped, basin of Härsvatten. This is consistent with the suggestion that allochthonous OM is recalcitrant to sediment mineralization after its degradation in the catchment and within the water column (Gudasz et al., 2012).

Overall, our molecular characterization of OM in the south basin suggests an enrichment in algal versus allochthonous OM (e.g., higher N-compounds:carbohydrates) in the deeper areas of a deep, simple lake basin, in line with previously reported sediment C:N ratios along lake-basin transects (Kumke et al., 2005; Dunn et al., 2008; Bruesewitz et al., 2012). Given our data on the degradation status of algal and allochthonous OM, we believe that this trend in OM quality results from preferential degradation of algal versus allochthonous OM in sediments at

shallower/intermediate water depth in addition to the known focusing of living, and residues of,
authochthonous OM towards deeper sites (Ostrovsky and Yacobi, 1999).

*3.3.2 Spatial variability in the central, north and east basins and near-shore locations*

In the northern half of the lake, 11 of 19 locations fall within $cluster_{geo}1$ (Fig. 1c), which
distinguishes itself geochemically only by somewhat lower than average concentrations of elements
often associated with (oxy)hydroxides (i.e., Fe, Mn, As, P and Co; Table 1 and Sect. 3.1.2).
Sediments from the shallowest locations can potentially fall in one of four different clusters
($clusters_{geo}1$, 3, 4 or 6). Thus, for the northern half of the lake there is no evidence of sediment
focusing. The effect is either limited by the more gentle slopes of the north and east basins (Blais
and Kalff, 1995), modified by the water circulation resulting from the prevailing winds towards the
north-east (Bindler et al. 2001, Abril et al., 2004), and/or interrupted by aquatic vegetation that acts
as a sediment trap (Benoy and Kalff, 1999). Aquatic vegetation represents a major source of OM to
the sediments of the north, east and central basins ($clusters_{OM}$ 1, 4 and 6; Fig. 1d; Table 1 and Sect.
3.2.2). The enrichment of aquatic higher plant or moss OM in these sediments is consistent with
field observations during the original sediment coring in winter 1997, where mosses and *Isoetes* (a
vascular angiosperm plant) were observed in some parts of the lake to a depth of at least 10 m
(Bindler et al., 2001). The presence of such submerged vegetation in Härsvatten is favored by its
acidic, clear water (i.e., deeper light penetration), as previously observed for other acidified boreal
Swedish lakes, such as the nearby lake Gårdsjön (Andersson, 1985; Grahn, 1985). Benthic aquatic
vegetation is also favored in the northern half of Härsvatten by the more gentle slopes,
comparatively shallow water depth and thus greater availability of light than in the deep, steeper-
sloped south basin where allochthonous input appears as the main source of higher plant and moss
OM (Sect. 3.2.2; Table 1).

The sediments found across the north and east basins and at the deeper sampling site of the central area (clusters$_{OM}$ 1; Fig. 1d) have the highest proportions of fresh, labile higher plant and moss OM, e.g. anhydrosugars (Sect. 3.2.2; Table 1). Also, the proportions of fresh, labile algal OM is as high as in the deeper anoxic sediments of the main south basin and two times higher than in the sediments found at shallow water depth in the south basin and central areas, although these sites span the same water depth range (3–11 m) and have relatively similar bSi contents (Table 1). These results indicate the accumulation of fresh autochthonous, both plant and algal, OM in sediments associated with in-lake vegetation even if they are below or within the epilimnion (i.e., supposed oxic conditions). A possible explanation is that the input of labile, decomposing in-lake higher plant and moss OM consumes oxygen and results in locally anoxic conditions in the sediment, which in turn lower OM mineralization rates (Bastviken et al., 2004; Isidorova et al., 2016). This hypothesis may explain the lower than whole-lake average concentrations of elements or elemental ratios often associated with (oxy)hydroxides (i.e., Fe, Mn, As, Co, P contents and Fe:Al) in these epilimnitic/metaliminitic sediments (cluster$_{geo}$ 1; Table 1). This interpretation is consistent with laboratory experiments, where, for example, Kleeberg, 2013 had shown that inputs of macrophyte residues to sediments results in oxygen depletion and microbially mediated reduction of Fe and Mn oxides. However, we cannot rule out that other factors, such as shallow groundwater discharges that are rich in (oxy)hydroxides or diagenetic processes that lead to Fe enrichment in sediments, can be involved in the higher concentrations of Fe, Mn and other elements known to be associated with Fe and Mn (oxy)hydroxides in the sediments of the south basin as compared to the sediments of the north and south basins.

The shallow sites located between the north and east basins and between the central area and the south basin (i.e., cluster$_{geo}$ 3 and clusters$_{OM}$ 4 and 6; Fig. 1c and 1d) have higher than whole-lake averages bSi contents and values for the ratio in-lake:terrestrial higher plant and moss OM, suggesting that these sediments receive plant OM from in-lake vegetation and algal OM from benthic production (Table 1). However, the proportions of fresh, labile plant and algal OM based on

N-compound and carbohydrate composition in these central sediments are much lower than in the
sediments found across the north and east basins (Table 1). Probably, these central areas are not
sites for aquatic vegetation growth, but receive in-lake plant OM produced within the north and east
basins that has been degraded during transport and/or is degraded at these shallow central sites due
to more oxic conditions. More oxic conditions at these shallow central sites are also suggested by a
higher occurrence of Fe and Mn (oxy)hydroxides (Fe, Mn, As, Co, and P contents, Fe:Al and
Mn:Fe above 10 % of whole-lake averages; Table 1). Among these shallow central sites, two
locations (cluster$_{OM}$ 6) are specifically rich in higher plant and moss lipids (i.e., C23-35:0; Table S3
in the SI) and, have high proportions of fresh higher plant OM based on lignin composition while
the proportions of fresh carbohydrates (anhydrosugars) versus total carbohydrates is low (Table 1).
This suggests preservation of higher plant cell-wall lipids and lignin with respect to carbohydrates
at these two shallow sites, in agreement with the known faster assimilation of carbohydrates versus
lipid and lignin structures (Bianchi and Canuel, 2011). However, no reasonable hypothesis could be
given to explain this difference in OM molecular composition between the sediments at sites M5-6
and the ones at sites N10 and M1 given their similar water depth and elemental geochemistry.

Among the sediments found in a small number of near-shore locations (cluster$_{geo}$ 4 and

cluster$_{OM}$ 5; n=4), three are located in two more-sheltered bays at the northwestern corner and the
southern end of the lake that are more protected from wind circulation (Bindler et al. 2001, Abril et
al. 2004). The sediments of these three locations predominantly accumulate terrestrial OM as
indicated by the abundance in lignin oligomers and the ratio indicative of in-lake:terrestrial plant
OM that are respectively above and below 10 % of the whole-lake averages (Table 3).
Accumulation of OM coming from the coniferous-forested catchment most probably explained the
high OM content (i.e. 52-58 %, which is as high as in the deeper sediments of the main south basin)
as well as the high concentrations of S and trace metals (i.e., Hg, Pb and Zn) in these near-shore
sediments (Table 1). Boreal forest soils are known to be enriched in S and trace metals because
their organic fraction retains atmospheric S and trace metals deposited over the industrial era
(Johansson and Tyler, 2001). Also, there is evidence that the transport of terrestrial OM to boreal
aquatic ecosystems is associated with significant inputs of trace metals (Grigal, 2002; Rydberg et
al., 2008). Alternatively, high S and trace metal contents could be due to accumulation of metal
sulfides due to near-shore groundwater gradients and/or anoxic conditions, or to redox cycling
related to the important input of terrestrial OM.

*3.3.3 Implication for in-lake and/or global elemental (e.g., C, nutrients, trace metals) cycling*

The molecular composition of natural OM has been shown to exert a strong influence on key
biogeochemical reactions involved in in-lake and global cycling of C, nutrients and trace metals,
such as C mineralization or nutrients/trace metals sorption and transformations into mobile and/or
bioavailable species (Drott A et al., 2007; Sobek et al., 2011; Gudasz et al., 2012; Tjerngren et al.,
2012; Kleeberg, 2013; Bravo et al., 2017). Our work demonstrates that OM molecular composition
can vary significantly within a single lake system in relation to basin morphometry, lake chemical
and biological status (e.g., presence of macrophytes, which is influenced by, e.g., acidification) and
the molecular structure/properties of the different OM compounds (e.g., higher resistant of
allochthonous versus autochthonous OM upon degradation). Our results further show that it may be
problematic to extrapolate data on OM composition from only a few sites or one basin when scaling
up to a whole lake. Thus, investigating sedimentary processes and the resulting fate of C and trace
elements using sampling strategies focused on the deepest area of a lake or on single transects from
shallower to deeper sites, may not fully capture the variation in either elemental geochemistry or
OM composition.
Overall, this study underlines that the OM molecular composition and its spatial heterogeneity
across a lake are two factors that should be considered to better constrain processes involved in the
fate of C, nutrients and trace metals in lake ecosystems, to improve whole-lake budgets for these
elements and to better assess pollution risks and the role of lakes in global elemental cycles.

**Data availability.**

The supporting information includes the raw data for sediment elemental geochemical parameters and for the 41 groups of organic compounds (resulting from the identification of 162 pyrolytic organic compounds) used for the statistical analysis and discussion. Raw data for the 162 pyrolytic organic compounds will be provided upon request from the authors.

**Author contribution.**

J. Tolu and R. Bindler designed research. J. Tolu performed Py-GC/MS analyses with help from L. Gerber and did the data treatment with the data processing pipeline of L. Gerber. J. Tolu and J. Rydberg performed XRF and mercury analyses. J. Tolu and C. Meyer-Jacob performed FTIR measurements and C. Meyer-Jacob determined the inferred bSi. J. Tolu, J. Rydberg, C. Meyer-Jacob and R. Bindler interpreted the data. J. Tolu prepared the manuscript with consistent contributions from J. Rydberg, R. Bindler and C. Meyer-Jacob.

**Acknowledgement**s.

We would like to thank the university of Umeå (Sweden) for the funding of this work, which was supported by the environment's chemistry research group as well as the Umeå plant science center for making the Py-GC/MS available to us and Junko Takahashi Schmidt for the technical support in the Py-GC/MS laboratory.

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

**Table 1.** Summary statistics for sediment elemental geochemistry

| | | Whole sample collection except the two outliers | | | | | Outliers (M4, S15) | | | |
|---|---|---|---|---|---|---|---|---|---|---|
| | Unit | Av.[a] ± sd[b] | CV[c] | Median | A:M[d] | Min[e]-Max[f] | Av. ± sd | CV[c] | Median | Min-Max |
| **W.D.** | m | **9** ± 7 | *74* | 7 | *1.23* | 2-25 | **3.4** ± 0.6 | *19* | 3 | 2.9-3.8 |
| **B.D.** | g cm$^{-3}$ | **0.06** ± 0.02 | *38* | 0.06 | *1.05* | 0.02-0.13 | **0.67** ± 0.09 | *14* | 0.06 | 0.61-0.74 |
| **bSi** | % | **13** ± 6 | *48* | 12 | *1.05* | 4-25 | **1.9** ± 0.2 | *0* | 12 | 1.7-2.0 |
| **LOI** | % | **38** ± 10 | *26* | 37 | *1.01* | 10-58 | **3.6** ± 0.8 | *20* | 37 | 3.0-4.1 |
| **[S]** | mg kg$^{-1}$ | **11876** ± 5920 | *50* | 11305 | *1.05* | 4685-29190 | **2570** ± 552 | *21* | 10610 | 2180-2960 |
| **[Br]** | mg kg$^{-1}$ | **149** ± 35 | *23* | 152 | *0.99* | 71-225 | **16** ± 7 | *44* | 148 | 11-21 |
| **[Cu]** | mg kg$^{-1}$ | **34** ± 13 | *37* | 32 | *1.07* | 12-75 | **9** ± 3 | *31* | 31 | 7-11 |
| **[Ni]** | mg kg$^{-1}$ | **19** ± 4 | *24* | 19 | *0.99* | 10-27 | **12** ± 4 | *35* | 19 | 9-15 |
| **[Hg]** | µg kg$^{-1}$ | **337** ± 202 | *60* | 286 | *1.18* | 117-1152 | **28** ± 9 | *33* | 274 | 21-34 |
| **[Pb]** | mg kg$^{-1}$ | **192** ± 74 | *39* | 184 | *1.05* | 58-422 | **22** ± 16 | *76* | 178 | 10-33 |
| **[Zn]** | mg kg$^{-1}$ | **219** ± 108 | *49* | 207 | *1.06* | 43-445 | **50** ± 16 | *31* | 200 | 39-61 |
| **[Al]** | % | **3** ± 1 | *17* | 3 | *1.06* | 2-4 | **5.67** ± 0.01 | *0.1* | 3 | 5.66-5.67 |
| **[Y]** | mg kg$^{-1}$ | **25** ± 8 | *32* | 25 | *1.01* | 7-43 | **20** ± 4 | *18* | 25 | 17-22 |
| **[Fe]** | % | **5** ± 3 | *65* | 4 | *1.26* | 1-12 | **3.4** ± 0.1 | *4* | 4 | 3.3-3.5 |
| **[As]** | mg kg$^{-1}$ | **35** ± 20 | *56* | 28 | *1.26* | 5-73 | <DL | | 27 | 0-0 |
| **[P]** | mg kg$^{-1}$ | **1624** ± 741 | *46* | 1401 | *1.16* | 655-3769 | **949** ± 57 | *6* | 1389 | 908-989 |
| **[Mn]** | mg kg$^{-1}$ | **729** ± 1690 | *232* | 180 | *4.06* | 94-7981 | **1060** ± 845 | *80* | 184 | 462-1657 |
| **[Co]** | mg kg$^{-1}$ | **19** ± 15 | *77* | 14 | *1.39* | 5-76 | **17** ± 9 | *56* | 14 | 10-23 |
| **[Ca]** | mg kg$^{-1}$ | **5261** ± 1306 | *25* | 5213 | *1.01* | 2860-9300 | **26540** ± 7566 | *29* | 5283 | 21190-31890 |
| **[K]** | mg kg$^{-1}$ | **4426** ± 1020 | *23* | 4485 | *0.99* | 2420-6140 | **10510** ± 2616 | *25* | 4580 | 8660-12360 |
| **[Mg]** | mg kg$^{-1}$ | **1488** ± 354 | *24* | 1500 | *0.99* | 870-2130 | **7495** ± 3599 | *48* | 1515 | 4950-10040 |
| **[Na]** | mg kg$^{-1}$ | **1795** ± 659 | *37* | 1743 | *1.03* | 440-3380 | **10695** ± 587 | *5* | 1783 | 10280-11110 |
| **[Si$_{inorganic}$]** | % | **11** ± 4 | *33* | 11 | *1.06* | 4-21 | **23** ± 1 | *3* | 11 | 22-23 |
| **[Sr]** | mg kg$^{-1}$ | **55** ± 16 | *29* | 55 | *1.01* | 27-116 | **235** ± 24 | *10* | 55 | 218-252 |
| **[Ti]** | mg kg$^{-1}$ | **2115** ± 495 | *23* | 2200 | *0.96* | 997-2870 | **4357** ± 2348 | *54* | 2215 | 2697-6017 |
| **[V]** | mg kg$^{-1}$ | **63** ± 15 | *23* | 60 | *1.05* | 36-101 | **75** ± 23 | *31* | 60 | 58-91 |
| **[Zr]** | mg kg$^{-1}$ | **101** ± 31 | *31* | 100 | *1.01* | 39-160 | **158** ± 6 | *4* | 103 | 153-162 |

[a] Av.: average; [b] sd: standard deviation; [c] CV: coefficient of variation calculated as relative standard deviation in %; [d] A:M: ratio between average and median; [e] Min.: minimal value; [f] Max.: maximal value

**Table 2.** Summary statistics for the molecular composition of sediment OM given as relative abundances (expressed in %) of the 41 groups of pyrolytic organic compounds, which belong to 13 classes of OM as indicated by the grey shading *(to be continued)*

| | Compounds included | Av[a] ± sd | CV | Median | A:M | Min-Max |
|---|---|---|---|---|---|---|
| **Carbohydrates** | | | | | | |
| (Alkyl)-furans & furanones | 3-furaldehyde, 2-furaldehyde, 2-acetyl-furan, Methyl-3-furaldehyde, 2(5H)-furanone, Methyl-2-furaldehyde, Dihydro-methyl-furanone, Methyl-2(5H)-furanone, Methyl-2-furaldehyde | **15** ± 4 | 30 | *14* | 1.06 | 8-28 |
| Hydroxy- or carboxy-furans & furanones | 2-Furancarboxylic acid, methyl ester; 2,5-Dimethyl-4-hydroxy-3(2H)-furanone; 5-(hydroxymethyl)-2-furaldehyde | **4.1** ± 1.2 | 29 | *4.0* | 1.03 | 0.8-7.5 |
| Pyrans | 5,6-dihydro-pyran-2-one, 4-hydroxy-5,6-dihydro-pyran-2-one | **3.4** ± 1 | 30 | *3.2* | 1.06 | 1.2-5.3 |
| Dianhydrorhamnose | Dianhydrorhamnose | **1.6** ± 0.5 | 28 | *1.7* | 0.99 | 0.3-2.7 |
| Levoglucosenone | Levoglucosenone | **2.2** ± 0.4 | 20 | *2.2* | 1.00 | 1.3-3.1 |
| Anhydrosugars | Anhydrohexose, Levogalactosan, Levomannosan, Levoglucosan | **3.7** ± 2.6 | 71 | *2.5* | 1.46 | 0.8-11 |
| **Chitin derived compounds** | | | | | | |
| Chitin-derived compounds | Acetamide, 3-acetamido-furan, 3-acetamido-4-pyrone, Oxazoline | **2.5** ± 1 | 40 | *2.6* | 0.98 | 0.2-4.2 |
| **N-compounds** | | | | | | |
| (Alkyl)pyridines | Pyridine, 2-methyl-pyridine, 3/4-methyl-pyridine | **0.3** ± 0.1 | 34 | *0.3* | 0.95 | 0.1-0.5 |
| Pyridines_O, *i.e. pyridines with side chain containing a "C=O" function* | 2-acetylpyridine, 3-acetylpyridine, 2-Methyl-5-acetoxypyridine | **0.7** ± 0.1 | 18 | *0.7* | 1.00 | 0.2-0.9 |
| (Alkyl)pyrroles | Pyrrole, Methyl-pyrrole | **2.4** ± 0.5 | 22 | *2.4* | 1.01 | 1.7-3.5 |
| Pyrroles_O , *i.e. pyrroles with side chain containing a "C=O" function* | 2-formyl-pyrrole, 2-acetyl-pyrrole, 2-formyl-1-methylpyrrole | **1.0** ± 0.2 | 25 | *0.9* | 1.04 | 0.5-1.4 |
| Pyrroledione & pyrrolidinedione | 2,5-pyrroledione, 2,5-pyrrolidinedione | **1.2** ± 0.3 | 29 | *1.2* | 0.98 | 0.2-1.7 |
| Aromatic N- compounds | Benzeneacetonitrile, Benzenepropanenitrile | **0.8** ± 0.3 | 36 | *0.8* | 1.03 | 0.3-1.4 |
| Indoles | Indole, Methyl-indole | **1.5** ± 0.4 | 24 | *1.5* | 1.03 | 0.5-3.1 |
| Diketodipyrrole | Diketodipyrrole | **0.8** ± 0.2 | 22 | *0.8* | 1.01 | 0.4-1.2 |
| Diketopiperazines | Pro-Ala, Pro-Val, Pro-Val, Cyclo-Leu-Pro, Pro-Pro, Pro-Phe | **1.5** ± 0.4 | 30 | *1.5* | 1.02 | 0.3-2.6 |
| Alkylamides | 6 alkylamides | **0.6** ± 0.3 | 51 | *0.6* | 1.06 | 0.1-1.7 |
| **Phenols** | | | | | | |
| Phenols | Phenol, 2-methyl-phenol, 3/4- methyl-phenol, dimethyl-phenol, Ethyl-phenol, Propenyl-phenol | **8** ± 1 | 15 | *8* | 1.02 | 4.4-11.4 |
| **Lignin** | | | | | | |
| Syringols | Syringol, 4-vinyl-syringol, 4-formyl-syringol, 4-allenesyringol, Acetosyringone | **0.5** ± 0.4 | 83 | *0.4* | 1.32 | 0.1-1.9 |
| Guaiacols | Guaiacol, Ethyl-guaiacol, 4-vinyl-guaiacol, 4-propenyl-guaiacol, Vanillin, 4-alleneguaiacol, Acetovanillone, Methyl-ester-vanillic acid, , Guaiacylacetone | **3.6** ± 2.3 | 65 | *2.9* | 1.24 | 1.1-13.5 |

**Table 2.** Summary statistics for the molecular composition of sediment OM given as the relative abundances (expressed in %) of the 41 groups of Py organic compounds, which belong to 13 classes of OM that are indicated by the grey shading *(following part)*

| | | | | | | |
|---|---|---|---|---|---|---|
| **Chlorophylls** | | | | | | |
| Pristenes | Prist-1-ene, Prist-2-ene | **2.7** ± 0.8 | 28 | *2.8* | 0.97 | 0.4-4.6 |
| Phytadienes | Phytadiene 1, Phytadiene 2 | **1.9** ± 0.7 | 35 | *1.8* | 1.04 | 0.2-3.6 |
| ***n*-alkenes** | | | | | | |
| C9-16:1 | *n*-alkenes C9, C13, C14, C16 | **3.5** ± 0.8 | 23 | *3.6* | 0.98 | 1.8-5.1 |
| C17-C22:1 | *n*-alkenes C17, C18, C19, C20, C21, C22 | **6** ± 1 | 17 | *6.2* | 0.97 | 3.5-8.9 |
| C23-26_1 | *n*-alkenes C23, C24, C25, C26 | **2.9** ± 0.9 | 32 | *2.7* | 1.09 | 0.6-5.4 |
| C27-28:1 | *n*-alkenes C27, C28 | **0.8** ± 0.4 | 47 | *0.7* | 1.10 | 0.1-1.4 |
| ***n*-alkanes** | | | | | | |
| C10-16:0 | *n*-alkanes C10, C11, C12, C13, C14, C15, C16 | **2.5** ± 0.6 | 23 | *2.5* | 1.03 | 1.3-4.1 |
| C17-22:0 | *n*-alkanes C17, C18, C19, C20, C21, C22 | **3.9** ± 0.8 | 21 | *4.0* | 0.98 | 1.6-5.4 |
| C23-26:0 | *n*-alkanes C23, C24, C25, C26 | **2.8** ± 1.4 | 49 | *2.7* | 1.07 | 1.4-8.8 |
| C27-35:0 | *n*-alkanes C27, C28, C29, C30, C31, C32, C33, C35 | **4.3** ± 3.5 | 80 | *3.6* | 1.20 | 1.1-21.3 |
| **Alkan-2-ones** | | | | | | |
| 2K C13-17 | Alkan-2-ones C13, 16, 17 | **1.3** ± 0.4 | 33 | *1.4* | 0.96 | 0.6-2.2 |
| 2K C19-21 | Alkan-2-ones C19, 20, 21 | **0.3** ± 0.1 | 45 | *0.3* | 0.97 | 0-0.8 |
| 2K C23-31 | Alkan-2-ones C23, 14, 25, 26, 27, 28, 29, 31 | **1.3** ± 0.8 | 62 | *1.1* | 1.24 | 0.1-3.3 |
| **Steroids** | | | | | | |
| Steroids | Cholest-2-ene, Cholesta-3,5-diene, Stigmasta-5,22-dien-3-ol, acetate, Sitosterol, Cholesta-3,5-dien-7-one, Stigmasta-3,5-dien-7-one | **1.2** ± 0.9 | 70 | *1.1* | 1.10 | 0-4.3 |
| **Tocopherols** | | | | | | |
| Tocopherols | γ-Tocopherol, α-Tocopherol | **0.3** ± 0.3 | 106 | *0.2* | 1.75 | 0-1.5 |
| **Hopanoids** | | | | | | |
| Hopanoids | Trinosphopane, Norhopene, 22,29,30-trisnorhop-17(21)-ene, 22,29,30-trisnorhop-16(17)-ene, Norhopane, 25-norhopene | **1.3** ± 0.4 | 31 | *1.4* | 0.94 | 0.2-1.9 |
| **(Poly)aromatics** | | | | | | |
| Benzene | Benzene | **0.9** ± 0.4 | 43 | *0.8* | 1.14 | 0.4-2.5 |
| Benzaldehyde | Benzaldehyde | **0.6** ± 0.3 | 41 | *0.6* | 1.08 | 0.3-1.5 |
| Acetylbenzene | Acetyl-benzene | **1.1** ± 0.4 | 39 | *1.0* | 1.10 | 0.6-2.3 |
| Alkylbenzenes C3-9 | Ethyl-methyl-benzene, Benzene C7, Benzene C9, | **1.9** ± 0.5 | 23 | *1.8* | 1.07 | 1.4-3.5 |
| Polyaromatics | Styrene, Indene, Dihydro-naphthalene, Dihydro-inden-1-one, 1-methyl-napthalene, 2-methyl-napthalene, Biphenyl, Fluorene, Anthracene | **1.4** ± 0.4 | 27 | *1.3* | 1.04 | 0.8-2.1 |

955

956

**Table 3.** Whole-lake and clusters average for a selection of elemental geochemical parameters and of ratios indicative of OM source types and their degradation status

**SPECIFIC FEATURES IN GEOCHEMISTRY**

| | Whole-lake[a] | Near-shore sites | North/East basins | South basin Shallower | South basin Intermediate depth | South basin Deeper | Shallow central areas |
|---|---|---|---|---|---|---|---|
| | | $Cluster_{geo}$ 4 | $Cluster_{geo}$ 1 | $Cluster_{geo}$ 6 | $Cluster_{geo}$ 2 | $Cluster_{geo}$ 5 | $Cluster_{geo}$ 3 |
| | (n[b]=42) | (n=4) | (n=13) | (n=10) | (n=8) | (n=3) | (n=4) |
| Water depth (m) | 9 ± 7 (78 %)[c] | 4 ± 2 | 5 ± 3 | 8 ± 3 | 15 ± 4 | 24 ± 1 | 2 ± 1 |
| Bulk density (g cm$^{-3}$) | 0.06 ± 0.02 (33 %) | 0.06 ± 0.03 | 0.07 ± 0.02 | 0.07 ± 0.02 | 0.05 ± 0.01 | 0.026 ± 0.009 | 0.10 ± 0.02 |
| [bSi] (%) | 13 ± 6 (46 %) | 12 ± 6 | 13 ± 3 | 15 ± 7 | 7 ± 3 | 4.2 ± 0.3 | 21 ± 4 |
| LOI] (%) | 38 ± 10 (26 %) | 50 ± 12 | 39 ± 5 | 34 ± 7 | 37 ± 4 | 52 ± 2 | 20 ± 8 |
| [S] (mg kg$^{-1}$) | 11876 ± 5920 (50 %) | 17510 ± 833 | 11683 ± 3440 | 7550 ± 1900 | 12896 ± 3315 | 26227 ± 4833 | 4879 ± 148 |
| [Br] (mg kg$^{-1}$) | 149 ± 35 (23 %) | 130 ± 6 | 153 ± 36 | 145 ± 35 | 154 ± 19 | 204 ± 26 | 116 ± 32 |
| [Cu] (mg kg$^{-1}$) | 34 ± 13 (38 %) | 36 ± 5 | 28 ± 6 | 30 ± 7 | 42 ± 6 | 65 ± 10 | 24 ± 13 |
| [Ni] (mg kg$^{-1}$) | 19 ± 5 (25 %) | 21 ± 1 | 18 ± 4 | 17 ± 2 | 21 ± 4 | 27 ± 1 | 12 ± 4 |
| [Hg] (μg kg$^{-1}$) | 337 ± 202 (60 %) | 407 ± 141 | 251 ± 47 | 230 ± 69 | 427 ± 94 | 917 ± 212 | 203 ± 87 |
| [Zn] (mg kg$^{-1}$) | 219 ± 108 (49 %) | 279 ± 31 | 212 ± 68 | 139 ± 42 | 305 ± 86 | 417 ± 33 | 63 ± 16 |
| [Fe] (%) | 5 ± 3 (60 %) | 3.1 ± 2.1 | 2.7 ± 1.7 | 3.6 ± 1.5 | 9.1 ± 2.4 | 4.3 ± 2.2 | 5.5 ± 1.7 |
| Fe:Al | 1.5 ± 0.8 (53 %) | 1.0 ± 0.5 | 1.0 ± 0.6 | 1.1 ± 0.3 | 2.5 ± 0.9 | 1.3 ± 0.6 | 1.9 ± 0.3 |
| [As] (mg kg$^{-1}$) | 35 ± 20 (57 %) | 27 ± 17 | 26 ± 16 | 25 ± 11 | 64 ± 11 | 48 ± 14 | 29 ± 9 |
| [P] (mg kg$^{-1}$) | 1624 ± 741 (46 %) | 927 ± 240 | 1065 ± 295 | 2088 ± 730 | 2074 ± 275 | 2766 ± 869 | 1224 ± 216 |
| [Mn] (mg kg$^{-1}$) | 729 ± 1690 (231 %) | 162 ± 53 | 182 ± 67 | 184 ± 50 | 305 ± 93 | 171 ± 13 | 5700 ± 1597 |
| Mn:Fe | 0.02 ± 0.03 (150 %) | 0.007 ± 0.002 | 0.008 ± 0.003 | 0.006 ± 0.002 | 0.004 ± 0.001 | 0.005 ± 0.002 | 0.111 ± 0.051 |
| [Co] (mg kg$^{-1}$) | 19 ± 15 (79 %) | 15 ± 8 | 12 ± 6 | 13 ± 5 | 26 ± 11 | 14 ± 2 | 49 ± 24 |
| [Pb] (mg kg$^{-1}$) | 192 ± 90 (47 %) | 199 ± 58 | 132 ± 53 | 115 ± 42 | 300 ± 59 | 315 ± 7 | 182 ± 96 |

**SPECIFIC FEATURES IN OM COMPOSITION**

| | | Whole-lake | Near-shore sites | North/East basins | South basin Shallower/intermediate depth | South basin Deeper | Shallow central areas | Shallow central areas |
|---|---|---|---|---|---|---|---|---|
| | | | $Cluster_{OM}$ 5 | $Cluster_{OM}$ 1 | $Cluster_{OM}$ 3 | $Cluster\_{OM}$ 2 | $Cluster_{OM}$ 4 | $Cluster_{OM}$ 6 |
| | | (n=42) | (n=4) | (n=16) | (n=14) | (n=3) | (n=3) | (n=2) |
| Water depth (W.D.) | | 9 ± 7 (78 %) | 4 ± 2 | 7 ± 5 | 11 ± 5 | 24.1 ± 0.5 | 3.2 ± 0.9 | 1.8 ± 0.1 |
| LOI (%) | | 38 ± 10 (26 %) | 50 ± 12 | 39 ± 4 | 36 ± 5 | 52 ± 2 | 24 ± 4 | 14 ± 6 |
| (C23-35:0+2K C23-31): Lignin[d] | In-lake:Terrestrial plant OM | 2 ± 1 (50 %) | 0.8 ± 0.5 | 3 ± 1 | 1.7 ± 0.4 | 1.8 ± 0.6 | 3 ± 1 | 19 ± 11 |
| N-compounds : Carbohydrates | Algal:Plant OM | 0.37 ± 0.09 (24 %) | 0.32 ± 0.08 | 0.35 ± 0.04 | 0.39 ± 0.05 | 0.6 ± 0.1 | 0.29 ± 0.02 | 0.23 ± 0.05 |
| Chlorophylls : Plant lipids+lignin | Algal:Plant OM | 0.18 ± 0.09 (50 %) | 0.10 ± 0.05 | 0.13 ± 0.06 | 0.24 ± 0.08 | 0.31 ± 0.07 | 0.18 ± 0.05 | 0.03 ± 0.03 |
| Proteins:(alkyl)pyrroles+ (alkyl)pyridines+Aromatic N | Algal OM (N-compounds) freshness | 0.3 ± 0.1 (33 %) | 0.39 ± 0.09 | 0.36 ± 0.05 | 0.22 ± 0.06 | 0.42 ± 0.06 | 0.20 ± 0.08 | 0.13 ± 0.08 |
| Phytadienes:pristenes[c] | Algal OM (chlorophylls) freshness | 0.4 ± 0.1 (25 %) | 0.4 ± 0.1 | 0.37 ± 0.09 | 0.40 ± 0.06 | 0.56 ± 0.05 | 0.42 ± 0.07 | 0.5 ± 0.2 |
| Anhydrosugars:(alkyl)furans & furanones | Plant OM (carbohydrates) freshness | 0.2 ± 0.2 (100 %) | 0.4 ± 0.2 | 0.3 ± 0.2 | 0.12 ± 0.11 | 0.14 ± 0.04 | 0.08 ± 0.01 | 0.042 ± 0.002 |
| Guaiacyl-acid:Guaiacyl-aldehyde[e] | Plant OM (lignin) freshness | 0.07 ± 0.03 (43 %) | 0.13 ± 0.02 | 0.07 ± 0.03 | 0.05 ± 0.02 | 0.04 ± 0.01 | 0.04 ± 0.03 | 0.10 ± 0.06 |
| Guaiacyl -2C: Guaiacyl -1C[e] | Plant OM (lignin) freshness | 0.8 ± 0.3 (38 %) | 1.23 ± 0.07 | 1.0 ± 0.2 | 0.5 ± 0.2 | 0.6 ± 0.2 | 0.5 ± 0.1 | 1.1 ± 0.2 |
| Syringyl-2C:Syringyl-1C[e] | Plant OM (lignin) freshness | 1.0 ± 0.8 (80 %) | 2.4 ± 0.3 | 1.1 ± 0.6 | 0.5 ± 0.2 | 0.6 ± 0.1 | 0.3 ± 0.3 | 1.4 ± 0.8 |

[a] whole-lake: averages of all analyzed sediment samples excluding the two outlier samples (sites M4, S15; cf. Sect. 3.1.1); [b] n: number of samples; [c] the data are presented as follow: **average** ± standard deviation *(relative standard deviation)*; [d] the compounds included in the ratios are given in detail in Table 2 and S1 in the SI.

Light grey background denotes average values below whole-lake average (<10 %); No background denotes values close to whole-lake average (±10 %); Dark grey background are values above whole-lake average (>10 %).

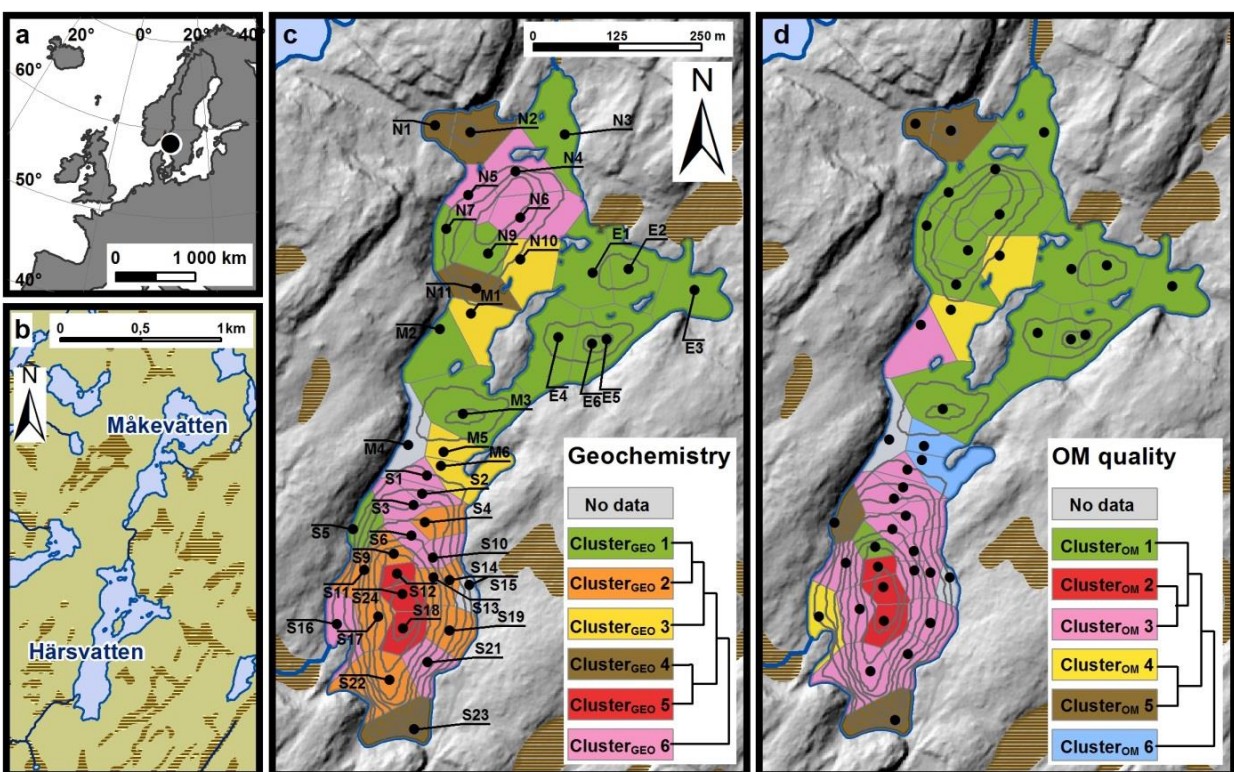

**Fig. 1** Maps of Härsvatten showing (a) its location in Europe; (b) its catchment with lakes, mires and larger streams; and (c, d) its bathymetry along with the spatial distribution of the 44 sampling sites and the six selected clusters based on sediment elemental geochemistry (c) and sediment OM molecular composition (d). In the panel c) and d), the dendrogram shows the relationship between the six identified clusters.

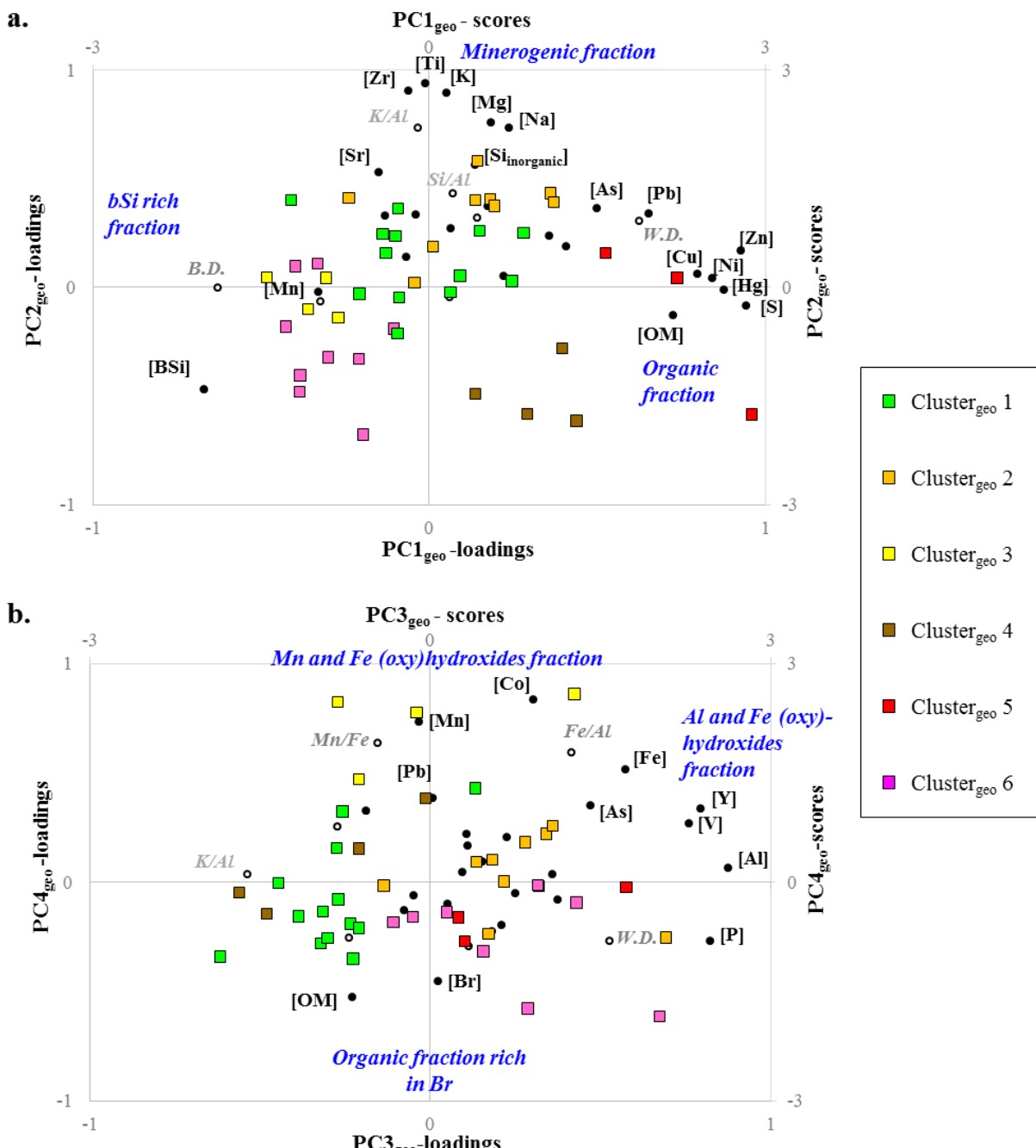

**Fig. 2** Combined loading- and score-plots for PCs 1-4 of the elemental geochemistry dataset. For the PC-loadings, filled circles correspond to active variables. Others variables (empty circle and italics letter) were added passively. Sediment samples are colored according to the results of the cluster analysis.

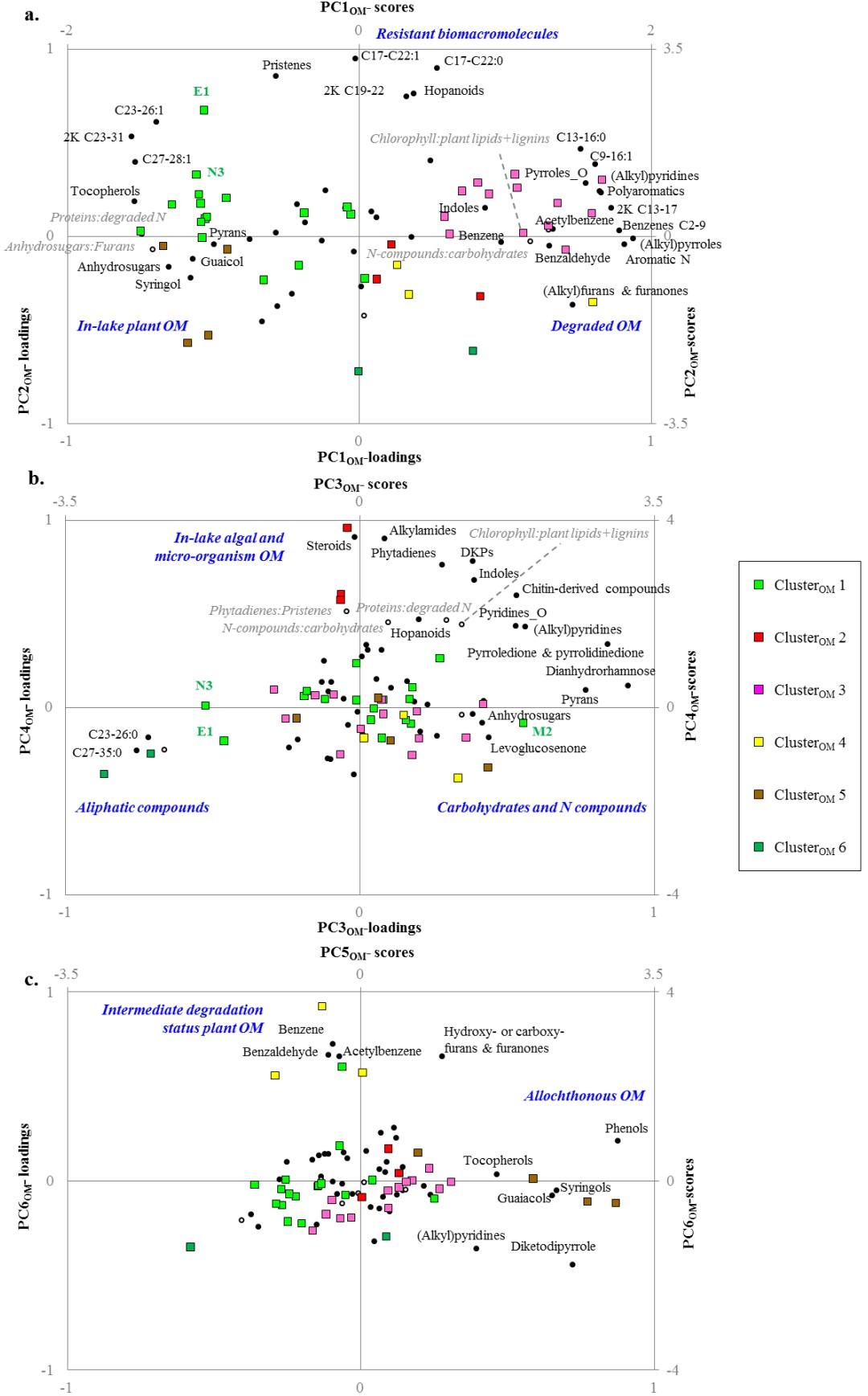

**Fig. 3** Combined loading- and score-plots for PCs 1-6 (a, b and c) of the OM molecular composition dataset (i.e. the 41 groups of organic compounds as defined in Table 2). For the PC-loadings, filled circles correspond to active variables. Others variables (empty circle and italics letter) were added passively. Sediment samples are colored according to the results of the cluster analysis.