# Peer review of "Spatial variability of organic matter molecular composition and elemental geochemistry in surface sediments of a small boreal Swedish lake"

_Biogeosciences, 2016_

## Referee Comment (RC1) · Anonymous Referee #1 · 7 Nov 2016

General Comments

This study explores the spatial variability in sediment geochemistry of a small boreal lake in order to better understand the processes that influence the concentrations of organic and inorganic constituents across a lake bottom. It analyzes a large array of surface sediment samples for major and trace elements and the molecular composition of organic matter (by pyrolysis-GC-MS) and uses standard multivariate methods (PCA, cluster analysis) to summarize a spatially coherent set of sedimentary "facies" that demonstrate correspondence between the organic and inorganic constituents. The results are interpreted in terms of organic matter sources (algae, aquatic plants, terrestrial plants), physical transport within the basin (focusing) and chemical transforma-

tion (decomposition, redox cycling) within the lake. In the deeper main basin of the lake, focusing is a dominant driver (as would be expected), although differences in OM mineralization between shallow (oxygenated) and profundal regions are an important overprint on algal OM composition. In a separate and shallower subbasin, there is less spatial differentiation in sediment quality, which is attributed to inhibition of focusing by macrophytes that dominate the OM molecular signal in this part of the lake.

Overall this is a fine study that builds on previous investigations of sedimentary processes in lakes. Its major contribution is the addition of detailed OM signatures that reveal both local provenance and degradation processes as well as a secondary influence on the distribution of inorganic (elemental) constituents. Interpretations are largely sound, though they are mostly descriptive explanations for the observed patterns. The larger implication of the paper is that care must be taken in using a single (or few) sediment sample(s) to characterize sediment composition of an entire lake. In itself, this is not a particularly novel idea, but is rather well documented in this careful and comprehensive study. One might have hoped to see a quatititvie exploration of how far off the mark would be a single sediment sample from the "deep hole" – as might be sampled in a more typical lake survey or paleolimnological study.

Specific Comments

(1) The title does not mention "sediments", which is a key word in the study, and the term "whole lake" is superfluous; suggested rewording: "Spatial variability of organic matter molecular composition and elemental geochemistry of sediments in a small boreal Swedish lake".

(2) A short methods description for sample collection and processing is needed.

(3) In several places in the results section, the molecular signatures of higher (or terrestrial) sourced organic matter is attributed simply to "plants" (e.g. lines 279, 380, 465, 466, 503). To some readers, this might also mean algae or aquatic macrophytes. Suggest changing to "terrestrial plants" or (if including macrophytes) "higher plants".

(4) In lines 438-441 an interesting trend of decreasing bSi from shallow to deep is attributed to the predominance of benthic diatom production in this clear, acidified lake. This is a reasonable argument, but it stops short of saying that there is lower diatom production in off-shore regions, presumably because of light limitation with increasing water depth.

(5) In lines 449-453 the depth-linked trend in algal OM degradation is attributed to greater exposure of shallow-water sediments to oxygen (as compared to profundal sediments). However, there could be another factor at play here – higher rates of sediment burial in deeper regions owing to focusing, which would also enhance preservation.

(6) Redox conditions are used to explain (in part) the distribution of Fe, Mn, P, etc., which is quite reasonable, as far as it goes. But much of the amorphous Fe and Mn entering lakes is delivered via shallow groundwater discharge, and it is not uncommon to find Fe enriched surface sediments in areas that such waters are discharged. Also Fe and Mn enrichment can be enhanced by diagenesis and diffusion within the sediment column. The discussion of redox elements needs to take these other processes into account.

(7) Elemental geochemistry was analyzed by WD-XRF, and as such represents bulk properties of the sediment. This is fine for elements that are largely confined to the silicate fraction of the sediment (e.g. Mg, Na, K), but can be misleading when there are multiple phases with different provenance – which is especially so for Fe, Mn, Ca, and sometimes Al. While most interpretation of these "mixed-phase" elements seem correct, there should be some mention of the fact that Fe in particular could reflect both transport of detrital material and solution transport of dissolved or amorphous phases.

(8) In lines 536-541 the high concentrations of S along with trace metals, Hg, Pb, Zn are ascribed to the accumulation of atmospheric pollutants in sheltered bays "...more protected from wind circulation". This seems like a pretty weak argument in that it invokes very localized deposition for which there is not much evidence or theoretical

mechanism. An alternative explanation is that the local enrichment is a consequence of the preferential accumulation of metal sulfides – for reasons related to redox cycling or near-shore groundwater gradients.

Technical Corrections

(line 1) Change to read: "The composition of sediment organic matter . . ."

(line 75) Change to read: "Beyond the rapidity of analysis and . . ." (delete "in terms')

(lines 84-85) Change to read: ". . . which factors or processes (e.g. provenance, transport pathway, mineralization). . ."

(line 93) Change to read: "This culturally acidified, clear-water . . ."

(line 101) Change "which" to "that"

(line 153) Insert "sediments" after "Härsvatten"

(liner 279) Change "On the contrary" to "In contrast"

(line 308) Change "readily assimilated" to "readily mineralized"

(lines 331-332) Change to read: ". . . are resitant to degradation."

(line 457) Change "bottoms" to "zones" or "regions"

(line 459) Change "south basin are" to "south basin is"

―――――――――――――――

---

## Referee Comment (RC2) · Anonymous Referee #2 · 27 Nov 2016

A spatial geochemical study of a lacustrine environments is original and represents a largely unexplored aspect of biogeochemistry. Most studies reach directly to the coring equipment and retrieve a core which then becomes the focus of the study assuming we understand all those spatial details. Although the findings are perhaps not that surprising this is a valuable contribution which shows how OM and inorganic geochemistry varies in a relatively small but complex lake. These findings will stimulate thinking about the processes involved in other lake systems, but the paper could have had more impact if this concept would have been developed a bit more. Why not show a few sediment profiles from this lake in relation to these findings and show the implications. That would be really interesting and relatively easy to add. Now it remains

just a hypothetical case. I have some criticism on how the results of the Py-GC/MS have been treated and would like to see some improvements made to the data that is derived from the Py-GC/MS.

Firstly, the title suggests that this study is characterising an entire lake. This is a bit misleading; I suggest to reword this to reflect more accurately the exact nature of this study.

The opening line of the abstract states: "The composition of organic matter (OM) exerts a strong control on biogeochemical processes in lakes, such as for carbon, nutrients and trace metals". Carbon, nutrients and trace metals are not processes... I suggest to reword this.

L8 no hyphen between pyrolysis and gas

L15 no hyphen between source and pools

L24 Py-GC-MS = Py-GC/MS this is the accepted convention of abbreviated this technique.

L45 delete: C being the main constituent of OM (I think that is obvious) L65 "little work has been done to detail the compositional variability of the OM matrix as a whole". Whilst I agree that in terms of spatial variation this is a virtually unknown territory, variation of the OM matrix in general is what many in the organic geochemistry community deal with. As such this is an overstatement and I believe the manuscript is doing an injustice to the many studies that have gone before. In Tolu et 2015 a more robust overview is given of previous Py-GC/MS studies. I would like to see more such acknowledgement and brief detail about previous achievements on OM matter composition studies.

Method section. Pyrolysis GCMS is traditionally carried out at 650 degrees or there about and 450 is significantly lower than the conventional methods. Since this deviation may have consequences, more detail should be given here. I read Tolu et al 2015 which

is a very interesting paper. But I am not sure if for example it can be said that aliphatic signal in lacustrine sediments, that is often observed at 650 degrees can be dismissed as being an artefact of the pyrolysis process. Previous work from others has clearly shown that this is to be expected when algenan is present, there is no question about the fact that the aliphatic signal is an important constituent of lacustrine OM... The fact that this particular signal significantly lowers at 450 concerns me. As such, it may be that lowering the temperature to 450 may cause bias toward the terrestrial components. Have you considered an alternative approach such as TMAH pyrolysis? Also it is not clear to me if the authors can distinguish between thermal desorption and pyrolysis. A lot of the compounds listed could also be explained as thermally extracted and may not be pyrolysis products at al (this might also explain the difference between 650 and 450).

L185 "Because these sediment samples also contained too little OM for Py-GC-MS analysis." Py-GCMS normally should be able to detect extremely low amounts of OM even with the small sample size that is described here, I find it hard to believe that nothing could be detected... give the %TOC of these samples in the discussion. Is it really nothing or were the results unexpected? This might also be a 450 vs 650 problem. I would imagine there is more complex OM than material that can thermally desorb. Given the effort that has been put into analysing the geochemical data I think it would have been better to determine the %TOC using elemental analysis as was done in Tolu et al 2015. LOI is not a very precise way to determine TOC. the pyrolysis results could be expressed per TOC.

Table 2: Give references to the information that is underlying the grouping. This table looks a bit rushed. The compound classes are not always clear and confusingly grouped. I object against the terminology used to group the compounds and the way how the grouping is organised. The authors must decide how to group the compounds and stick to a very clear well explained system of doing so. What are "Big furans"? please improve. Maleimie = Maleimide Proteins as a class of pyrolysis compounds is

not possible, I suspect protein derived is meant here. Lipids is the wrong term other compounds in the list are lipids too (eg hopanoids). Aliphatics is a better label.

Steroids and hopanoids have different biological origins, for a statistical approach such as what is presented here it would make a lot of sense to keep these separated. This would help with statistical analysis. A term such as lignin has a clear origin associated with it, a term such as "lipid" or "high mass compound" caries no such value.

The table as given in the supplement is much more useful. It is very important to see what contributes to these categories. Without it, the methodology becomes a black box. I suggest to combine table 2 and S1. This will avoid that the reader is wonders about what might be in a "big furan" category. I would suggest to rename some of the categories and rethink the groupings. This would mean that the PCA results are going to change.

I would like to see Retention Index values for each of the listed compounds in the table. This is very important as all compounds have been identified using the NIST database (according to the text) and this can be very unreliable. RI values together with the mass spectra (as have correctly been given) will give extra confidence. The mass spectra given are only a rough summary and do not have much value for a reader but with RI values it becomes much more solid. There are quite a few unknowns. I would like to challenge the authors: if they are not certain about the ID of a compound, how can you put a compound in a specific category?

---

## Author Comment (AC1) · 27 Dec 2016

We would like to thank referee 1 for the comments on our manuscript and agree on most of the mentioned suggestions. A point by point answer to the specific comments and the modifications we plan to do in the revised manuscript are given thereafter.

General comments.

We appreciate the comments pointing out that our data represent a "coherent set of sedimentary facies" and that "our interpretations are largely sounds". While a few studies have shown that OM quality of sediments can strongly vary across a single lake, our study was designed to determine what are the spatial patterns of a large set of or-

ganic compounds and compounds classes without discarding any lake location types (as pointed out by the referee, our study "analyses a large array of surface sediments samples"). Combined with elemental inorganic geochemistry data, we aimed at comprehensively investigating how OM molecular composition varies within the sediments of an entire lake basin and thereafter what factors and processes are involved in the spatial distribution of OM molecular composition (e.g., catchment sources, transport pathway, degradation. . .). Our study was not designed to get insights into specific mechanisms controlling the spatial distributions of sediments geochemistry and OM composition. Such an objective would require additional or others types of information (e.g., sediment ages, detailed catchment input) and a different sampling strategy, i.e. sediment sampling at higher spatial resolution and thus by focusing on some lake locations (e.g., the main south basin) to be a feasible research study. While our interpretations may thus appear as "mostly descriptive explanations for the observed patterns", our study clearly demonstrates that the spatial distribution of OM molecular composition is well explained by general/common factors and processes of lake ecosystems (i.e. sediment focusing, macrophytes, catchment input, mineralization). Our results give for the first time information on the OM molecular composition – as a whole matrix rather than only specific compounds or classes – that can be found in different locations of a lake in relation to such common factors. Hence, our study provide insights into the locations where it would be relevant to determine and compare the fate of C, nutrients and trace elements within a single lake, especially regarding the reactions for which OM molecular composition play a critical role, such as OC mineralization, mercury methylation, phosphorus mobility. As an example, our data indicate that the rates of OC mineralization and Hg methylation, two reactions strongly dependent on bacteria activity that are controlled by C and electron donor sources, should not only be investigated in the deeper sediments (rich in fresher algal OM) as often done, but also in, e.g., shallower sediments associated with macrophyte growth (rich in fresher higher plant and algal OM) as well as in the near-shore sheltered bays where there is accumulation of terrestrial OM including fresh carbohydrates. Thus, we believe that

our study is not restricted to "that care must be taken in using a single (or few) sediment sample(s) to characterize sediment composition of an entire lake". "Quantitative exploration of how far off the mark would be to study a single sediment sample from the deep hole – as might be sampled in a more typical lake survey or paleolimnological study." rather than studying multiple lake locations, has been done in previous studies focused on inorganic elements such as Hg or P to assess the errors made by calculating lake budgets or reconstructing past atmospheric pollution from data collected in the deeper sediments. However, it is difficult to apply such an approach to the OM molecular composition, which in our study includes 42 groups of organic compounds. Also, the aim of our study is not to show "that care must be taken in using a single (or few) sediment sample(s) to characterize sediment composition of an entire lake", but rather that for studies interested in assessing key biogeochemical processes related to, for example, carbon cycling or mercury methylation, care must be taken in scaling up to a whole lake basin from a limited number of sites. It is fully possible that a limited sampling strategy might overlook critical sites. Key aspects of our study are to: i) highlight that OM molecular composition, a key parameter of reactions controlling the fate of element of major concern, spatially vary and thus not only between-lake, but also within-lake, spatial variability of those reactions should be investigated and ii) provide a baseline work/ detailed information for the relevant lake locations to study.

Specific comments.

(1) Rewording the title

We agree with referee 1 that "sediments" is a key word of our study, and thus the title of our manuscript will be changed to "Spatial variability of organic matter molecular composition and elemental geochemistry of sediments in a small boreal Swedish lake".

(2) A short methods description for sample collection and processing is needed.

The 44 studied surface sediments (0-10cm) samples were collected as follow: "Short sediment cores (0-25 cm) were taken with a gravity corer from the ice-covered lake

in winter 1997 and 1998, and were sectioned into an upper sample (0-10 cm) and a lower sample (10-25 cm; not studied here) on-site. In the laboratory, the samples were weighed, freeze-dried, and reweighed to determine the water content and dry mass of the sediment. The freeze-dried samples have been stored in plastic containers within closed boxes shielded from light and at room temperature since winter 1997-1998. Before further analysis in this study, the samples were finely ground at 30 Hz for 3 min using a stainless steel Retsch swing mill." This paragraph which includes additional information about the sample collection and processing will replace the third paragraph in the sub-section "2.1 Study site and samples".

(3) Better clarity on OM sources

In order to avoid confusion with the term "plants", we will clarify this term by using instead "higher plants and mosses" or "higher plant" in the revised manuscript.

(4) Discussion of the decreasing bSi with depth (lines 438-441)

We agree with the reviewer that the lower bSi content in deeper than in shallower sediments is related to light limitation with increasing water depth. For us, this was meant by our explanation that diatom production in this clear, acidified lake is mostly benthic. For better clarity, we will replace the corresponding paragraph (lines 438-441 in the initial manuscript) by: "The bSi decline, from ∼15 to 4 %, indicates a decrease of diatom production with depth due to increasing light attenuation and thus suggests that the diatom assemblage is dominated by benthic species as shown for many acidified lakes, such as surrounding lakes in Svartedalen (e.g., Andersson, 1985; Anderson and Renberg, 1992)."

(5) Discussion of the depth-linked trend in algal OM degradation (lines 449-453)

We agree that higher rate of sediment burial can favor OM preservation (although the sedimentation rates in the deeper basins of Härsvatten shown in studies of past atmospheric Pb pollution are nonetheless very low with >500 yr for the upper 30 cm). Thus

higher mineralization of algal OM in the shallow and intermediate water depth sediments with respect to the deeper sediments could be related to lower accumulation rates in addition to oxic conditions, this latter hypothesis being supported by the elemental inorganic geochemistry. We will thus modify the corresponding sentence (lines 449-453 in the initial manuscript) as follows: "Although our results are based on the top 10 cm of sediment and thus account for different sediment ages, we suggest that the higher proportions of decomposed algal material, based on N-compound and chlorophyll composition (Table 1), at shallower and intermediate water depths (<21m) than at the deepest sites (23.5–24.5 m) reflect higher mineralization rates of OM in shallow/intermediate areas. Higher OM mineralization rates in these areas than at deeper sites are most probably due to more oxic conditions and lower sedimentation rates (i.e. longer exposition time of sediment OM at the sediment-water interface)."

(6) and (7) Discussion of the spatial distribution of Fe, Mn and Al

We fully agree with the reviewer that the spatial distribution of Fe, Al and Mn contents in lakes, including our study lake, is not restricted to the formation and/or preservation of Fe, Al and Mn (oxy)hydroxides, but depends on others factors and processes such as the discharge of groundwater, diffusion in the sediment or sediment diagenesis (comment 6). However, the main aim of our study was not to describe and comprehensively understand the spatial distribution of the sediment inorganic fraction, for which there are a number of published studies addressing this. Our study was focused on the spatial distribution of the sediment OM molecular composition as clearly stated in the introduction. Thus, we did not, and do not want to, discuss in detail the spatial distribution of the inorganic elements because we know the limitations of using only elemental contents to describe inorganic geochemistry, and more specifically that Fe, Al and Mn are not confined only to one mineral phase and they may reflect both detrital material and/or dissolved or amorphous phases (as pointed out by the referee 1 in comment 7). The main/more significant trends for Fe, Al and Mn highlighted by the PCA and cluster analyses are i) Fe, Al contents together with As, P and Fe:Al,

indicative of Fe, Al (oxy)hydroxides, are high or low in certain sediment locations and ii) Mn content together with Co, Pb and Mn:Fe, indicative of Mn (oxy)hydroxides are high in certain sediment locations. Hence, we use this information to discuss the relationship between OM composition and redox conditions which have evidences from others sediments parameters (such as sediment depth, epilimnion depth and/or sediment bottom type), have support from previous studies and appeared to the referee 1 as "quite reasonable, as far as it goes" (cf. comment 6). Therefore, in the revised manuscript, we agree to follow the suggestion of comment 7 and add information on the limitations of using Fe, Al and Mn contents to describe inorganic geochemistry in the sub-section "3.1.2. Principal components of the elemental geochemistry" where we present and interpret PCA results (lines 229-237 in the initial manuscript), as follows (the new sentences are in bold): "Positive loadings on PC3geo, which explains 16 % of the total variance, are found for Al and Fe along with As, P and Y (Fig. 2b). Compared to elements such as Mg, Na and K that are mostly confined to the silicate fraction of sediments, Fe and Al may reflect both detrital material and dissolved or amorphous phases. However, the fact that As and P contents as well as Fe:Al ratio plot with Fe and Al contents on the positive side of PC3geo, but not S content, strongly suggest that sediments with high PC3geo scores are associated with higher content of Fe and Al (oxy)hydroxides known to strongly bind both As and P (Mucci et al., 2000; Plant et al., 2005; Zhu et al., 2013). PC4geo captures 12 % of the total variance and separates Mn, Co, Pb and to a lesser extent Fe (positive loadings) from OM and Br (negative loadings; Fig. 2b). Although Mn, as Fe and Al, is not confined to a specific mineral phase and can reflect both detrital or dissolved and amorphous phases, the positive loadings are interpreted as reflecting Mn (oxy)hydroxides, which bind Pb, especially when they contain cobalt (Co) (Yin et al., 2011). This interpretation is supported by the positive loadings on PC4geo of the ratio Mn:Fe, often used as a paleolimnological proxy for bottom water oxygenation (Naeher et al., 2013). The negative loadings could indicate a terrestrial OM fraction that is rich in Br (Leri and Myneni, 2012)." However, we prefer to not include further discussion on the factors and processes involved in

the spatial distribution of Fe, Al and Mn, because it would lead to more emphasis in our discussion on the spatial distribution of inorganic geochemistry, which is not our objective, and our data based on total concentrations are not precise enough to do so.

(8) High concentrations of S and trace metals, Hg, Pb, Zn in sheltered bays (lines 536-541).

We think that our explanation for the high concentrations of S along with trace metals (Hg, Pb, Zn) in the sediments of near-shore sheltered bays of Härsvatten (sites N1-2 and S23) has been misunderstood. Hence, we have clarified the corresponding paragraph (lines 536-541 in the initial manuscript). Indeed, we do not ascribe the high concentrations of S and trace metals to accumulation of atmospheric pollutants directly. As pointed by the referee, this would have "invoked very localized deposition for which there is not much evidence or theoretical mechanism". We ascribe the high S and trace metals contents to the accumulation of terrestrial organic matter derived from the coniferous-forested catchment in theses sediments (enrichment in lignin). Indeed, the OM fraction of boreal terrestrial ecosystems (i.e. forest soils and wetlands) is well known to be enriched in S and trace metals because it retains atmospheric S and trace metals since the industrial revolution (Johansson et al., 2001). Also, there are strong evidences that the transport of terrestrial OM to boreal aquatic ecosystems is associated to significant input of trace metals (Grigal et al., 2002; Rydberg et al., 2008). However, we cannot rule out that accumulation of metal sulfide due to specific redox conditions or near-shore groundwater gradients is involved in the accumulation of S, Hg, Pb and Zn in the sediments of two near-shore sheltered bays of Härsvatten. We have thus added these hypotheses in the manuscript.

The corresponding paragraph (lines 536-541 in the initial manuscript) will be thus re-written as follows: "The sediments found in a small number of near-shore loca-tions (clustergeo 4 and clusterOM 5; n=4), three of which being located in two more-sheltered bays at the northwestern corner and the southern end of the lake and thus more protected from wind circulation (Bindler et al. 2001, Abril et al. 2004), predominantly accumulate terrestrial OM as indicated by the abundance in lignin oligomers and the ratio indicative of in-lake:terrestrial plant OM that are respectively above and below 10% of whole-lake average (Table 3). Accumulation of OM coming from the coniferous-forested catchment most probably explained the high OM content (i.e. 52-58%, which is as high as in the deeper sediments of the main south basin) as well as the high concentrations of S and trace metals (i.e., Hg, Pb and Zn) in these near-shore sediments (Table 1). Indeed, boreal forest soils are known to be enriched in S and trace metals because their organic fraction retains atmospheric S and trace metals since the industrial era (Johansson and Tyler, 2001). Also, there are evidences that the transport of terrestrial OM to boreal aquatic ecosystems is associated to significant input of trace metals (Grigal et al., 2002; Rydberg et al., 2008). Alternatively, high S and trace metal contents could also be linked to accumulation of metal sulfides due to near-shore groundwater gradients and/or anoxic conditions or redox cycling related to the large input of terrestrial OM."

This new paragraph will bring two additional references in the revised manuscript, which are: Johansson, K. & Tyler, G. Impact of atmospheric long range transport of lead, mercury and cadmium on the Swedish forest environment. Water, Air Soil Pollut. 425 279–297 (2001). Grigal, D. F. Inputs and outputs of mercury from terrestrial watersheds: a review. Environ. Rev. 10, 1–39 (2002).

Technical corrections. All technical corrections have been considered.

Please also note the supplement to this comment:
http://www.biogeosciences-discuss.net/bg-2016-361/bg-2016-361-AC1-supplement.pdf

---

## Author Comment (AC2) · 29 Dec 2016

We thank referee 2 for reviewing our manuscript. The response to the different comments and the intended modification in the manuscript are detailed thereafter.

1. "A spatial geochemical study of a lacustrine environments is original and represents a largely unexplored aspect of biogeochemistry. Most studies reach directly to the coring equipment and retrieve a core which then becomes the focus of the study assuming we understand all those spatial details. Although the findings are perhaps not that surprising this is a valuable contribution which shows how OM and inorganic geo- chemistry varies in a relatively small but complex lake. These findings will stimulate thinking about the processes involved in other lake systems, but the paper could

have had more impact if this concept would have been developed a bit more. Why not show a few sediment profiles from this lake in relation to these findings and show the implications. That would be really interesting and relatively easy to add. Now it remains just a hypothetical case."

We appreciate the positive view the reviewer has on our exploration of the geochemical and molecular composition of the sediments of a whole-lake basin. While we agree sediment profiles would be interesting, we disagree that it would be appropriate here for two main reasons. First, including an evaluation of sediment cores and all that such an evaluation would entail are outside the context of our study objective focused on spatial patterns in OM molecular composition and how these relate to geochemical variations and in-lake processes. We think there is a novelty already in this focus, with our combination of the pyrolysis-GC/MS characterization of OM molecular composition with elemental geochemistry. Second, including a temporal perspective by analyzing downcore changes would entail a fundamentally different discussion and would be a paper in its own right. We are uncertain why the reviewer considers this a hypothetical case. We believe the spatial context here is valuable for process-oriented studies on, e.g., carbon or trace elements cycling, which we think at this point is the main implication of our work. Indeed, our study clearly demonstrates that the spatial distribution of OM molecular composition is well explained by general/common factors and processes of lake ecosystems (i.e. sediment focusing, macrophytes, catchment input, mineralization). In consequence, our results provide insights into the locations where it would be relevant to determine and compare the fate of C, nutrients and trace elements within a single lake, especially regarding the reactions for which OM molecular composition play a critical role, e.g., OC mineralization, mercury methylation, phosphorus mobility. While the practical steps in analyzing sediment core samples themselves may not be a complicated task, presenting and evaluating downcore changes would require assessing diagenetic changes within each core (cf. our methods paper, which used varved sediments; Tolu et al. 2015), as well as including a full assessment of how the sediment record(s) reflect environmental changes over the represented timeframe of each

core. For Härsvatten, this would include the effects of acidification on lake biota (e.g., diatoms; Renberg et al. 1993), effects of acidification on carbon cycling (e.g., declines in TOC; Rosén et al. 2011), long-term changes in land use (e.g., cultural alkalization over past c. 1000 yrs until c. AD 1900; Renberg et al. 1993, Rosén et al. 2011), influence of 2000 yrs of atmospheric pollution deposition (e.g., Bindler et al. 2001), changes in catchment vegetation (e.g., spruce immigration), etc. These would all be interesting concepts to examine with lake sediment records and would all have to be addressed even with short surface cores because of the low sediment accumulation in this lake. A 30-cm surface core from the deeper basins would comprise >500 yrs of environmental changes, which would imprint on the OM quality. But as we first comment, all of this is outside the scope of our spatial assessment in OM quality, which we think is particularly relevant for process-oriented studies such as organic carbon mineralization, trace metals and nutrients sorption and transformations.

2. "Firstly, the title suggests that this study is characterising an entire lake. This is a bit misleading; I suggest to reword this to reflect more accurately the exact nature of this study."

We think that the referee 2 found our title to be misleading because we did not include the word "sediments", which is a keyword of our study as pointed out by the referee 1. The title of the manuscript will be replaced by "Spatial variability of organic matter molecular composition and elemental geochemistry of sediments in a small boreal Swedish lake".

3. "The opening line of the abstract states: "The composition of organic matter (OM) exerts a strong control on biogeochemical processes in lakes, such as for carbon, nutrients and trace metals". Carbon, nutrients and trace metals are not processes... I suggest to reword this."

To avoid such confusion, we will reword the sentence as follows: "The composition of sediment organic matter (OM) exerts a strong control on biogeochemical processes in

lakes, such as those involved in the fate of carbon, nutrients and trace metals."

4. "L8 no hyphen between pyrolysis and gas", "L24 Py-GC-MS = Py-GC/MS this is the accepted convention of abbreviated this technique.", "L15 no hyphen between source and pools" and "L45 delete: C being the main constituent of OM (I think that is obvious)."

These small technical/editing corrections will be made in the revised manuscript.

5. "L65 "little work has been done to detail the compositional variability of the OM matrix as a whole". Whilst I agree that in terms of spatial variation this is a virtually unknown territory, variation of the OM matrix in general is what many in the organic geochemistry community deal with. As such this is an overstatement and I believe the manuscript is doing an injustice to the many studies that have gone before. In Tolu et 2015 a more robust overview is given of previous Py-GC/MS studies. I would like to see more such acknowledgement and brief detail about previous achievements on OM matter composition studies."

In this part of the sentence line 65 "little work has been done to detail the compositional variability of the OM matrix as a whole", we are actually referring to a lack of data on the in-lake spatial variability of OM composition at the molecular level, for which the referee agrees that it is unexplored territory. Indeed, this sentence starts on discussing what has been done in term of in-lake spatial variability of OM quality, i.e. "Although there have been a few studies where the spatial complexity in OM quality within a whole-lake basin has been assessed using infrared spectroscopy, which yields qualitative information on variations in OM quality". However, to avoid such confusion and risking to make injustice to the high number of studies that have gone before on sediment OM molecular composition with different objectives, we would like to re-write this part of the sentence line 65 as follows "little work has been done to detail how the molecular composition of the sediment OM matrix as a whole varies spatially within a single lake". Given the large number of studies looking at variability of OM molecular composition

in sediments within a wide number of goals and contexts (such as OM mineraliza-tion and redox conditions, lake trophic status, urban pollution) and using methods that provide different levels of molecular information (i.e., Py-GC/MS, specific liquid extrac-tions associated to LC/GC-MS or LC-FTICR-MS analyses), we do not see how we can give "brief details about previous achievements on OM composition" in sediments as asked by the referee 2. We believe that adding a paragraph on presenting previous achievements on OM composition in sediments would greatly extend the length of the introduction and would be out of the scope of the study focused on a comprehensive understanding of in-lake spatial variability and distribution of OM composition.

6. "Method section."

6.a. "Pyrolysis GCMS is traditionally carried out at 650 degrees or there about and 450 is significantly lower than the conventional methods. Since this deviation may have consequences, more detail should be given here. I read Tolu et al 2015 which is a very interesting paper. But I am not sure if for example it can be said that aliphatic signal in lacustrine sediments, that is often observed at 650 degrees can be dismissed as being an artefact of the pyrolysis process. Previous work from others has clearly shown that this is to be expected when algenan is present, there is no question about the fact that the aliphatic signal is an important constituent of lacustrine OM: : : The fact that this particular signal significantly lowers at 450 concerns me. As such, it may be that lowering the temperature to 450 may cause bias toward the terrestrial components."

Pyrolysis-GC/MS is indeed generally carried out in environmental sciences at 650°C where the sample mass used for the Py-GC/MS analysis is generally above 1 mg. We have demonstrated in our methodological development (Tolu et al., 2015), in which we compared specifically 450 and 650°C, that when using 200 $\mu$g of sediment, we lose many Py products of lignin oligomers and especially all Py products of syringyl lignin oligomers due to increase in secondary reactions with a pyrolysis at 650°C. We also lose Py products indicative of fresh, algal or higher plant and mosses OM, e.g., some levosugars that are Py products of fresh higher plant and moss polysaccharides

or cellulose, and some 2,5 diketopiperazines that are Py products of proteins. As referee 2 pointed out, lower proportions of aliphatic compounds were obtained with a pyrolysis at 450°C, most probably due to less efficient volatilization of refractory organic molecules, which may include bio-macromolecules such as algaenan, cutin, suberin that are know to form aliphatic compounds during Py-GC/MS and especially n-alkenes/n-alkanes doublets with a number of C comprised between 16 and 24 carbons. However, we do not understand why "lowering the temperature to 450 may cause bias toward the terrestrial components as compared to a Py at 650 degree." We would like to first underline that not only bio-macromolecules from algae (such as algaenan), but also bio-macromolecules from higher plant and moss (such as cutin and suberin) are known to contribute strongly to the aliphatic signal in Py-GC/MS. Secondly, at 450°C, we are still identifying the series of n-alkenes/n-alkanes doublets characteristic of these bio-macromolecules and, actually, there is one principal component from the PCA analysis (i.e. PC2OM; Fig. 3a in the manuscript) which is associated with the n-alkenes/n-alkenes doublets with 17 to 22 carbon number and others organic compounds known for their recalcitrance. On the other hand, we believe that a pyrolysis at 650°C may cause a bias toward the refractory components of sediments OM (loss of many Py products from labile/fresh organic compounds such as polysaccharides and proteins), while this fraction of sediment OM is of much less relevance for process-oriented studies, which are one of the main implication of our study. We would prefer to not extend our manuscript by discussing details about the differences between pyrolysis at 450 and 650°C, given that Py at 650 °C was not carried out in this study and that this was already discussed in our paper presenting the methodological development where we compared results from the two treatments (Tolu et al., 2015).

6.b "Have you considered an alternative approach such as TMAH pyrolysis?"

We have considered, and even briefly tested, TMAH pyrolysis when we have performed our methodological development. The main differences between TMAH pyrolysis and regular pyrolysis is that it enables to identify more accurately acid protons containing

compounds, i.e. carboxylic acids and alcohols. However, this method has major draw-backs: 1. TMAH pyrolysis has been shown to be associated with really poor data repro-ducibility because the methylation process is non-exhaustive owing to steric hindrance or too large quantity of the original compounds (Chiavari et al. 1994). During our test (non pubished) with TMAH pyrolysis, we could clearly see very low reproducibility in the peak areas of compounds of interest (i.e. carboxylic acids). 2. We cannot identify the compounds that elute during the first 3-5 min due to solvent/TMAH elution. This part of the chromatograms contained most of the (alkyl)furans and (alkyl)pyrroles/pyridines, which are used as indicator of degraded OM by comparison with the levosugars (Py products of plant polysaccharides and cellulose) or the 2,5 diketopiperazines (Py prod-ucts of proteins). Hence, TMAH pyrolysis is a complementary analysis to regular py-rolysis for identifying the acid fraction of OM, but if one method has to be chosen, the regular pyrolysis is for us the most relevant one. We agree that providing data on the spatial variability of the abundance of the acid fraction of OM would be interesting. However, we believe they would not bring more insights on the factors and processes involved in the spatial distribution of OM composition than what we are discussing with our data from regular pyrolysis. Indeed, the acid fraction would most probably only show variations in the sources and degradation status of the sedimentary OM across Härsvatten, which are already clearly observed from the data obtained with regular py-rolysis. On the other hand, we would have to present the full dataset obtained by TMAH pyrolysis and the results of PCA and cluster analyses made on this specific dataset, which means it would significantly extend the length of our manuscript.

6.c. "Also it is not clear to me if the authors can distinguish between thermal desorption and pyrolysis. A lot of the compounds listed could also be explained as thermally extracted and may not be pyrolysis products at al (this might also explain the difference between 650 and 450)."

In Py-GC/MS, the sample reaches the Py oven which is already at 450°C and the volatilized fraction is injected in the GC/MS in a few seconds. This is technically

very different from thermodesorption-GC/MS analysis. May be, a part of the identified compounds that are volatilized during pyrolysis could also be released during thermodesorption analysis. However, at our knowledge, there is no way to identify simultaneously to Py-GC/MS what are those compounds, but it would require to run separately thermodesorption-GC/MS analyses. Moreover, we do not see how knowing whether the compound is directly released or is a "real" Py product will enable us to get more insights in our study because the most significant trends highlighted by the statistical analyses and by the comparison with the elemental geochemistry are about OM sources (autochthonous algal vs autochthonous higher plant and moss versus terrestrial OM) and degradation status, and are thus based on information about the structure, the origin and the reactivity of the identified organic compounds.

7. "L185 "Because these sediment samples also contained too little OM for Py-GC-MS analysis." Py-GCMS normally should be able to detect extremely low amounts of OM even with the small sample size that is described here, I find it hard to believe that nothing could be detected...give the %TOC of these samples in the discussion. Is it really nothing or were the results unexpected? This might also be a 450 vs 650 problem. I would imagine there is more complex OM than material that can thermally desorb."

Our explanation for the fact that we have no Py data for the sediments at sites M4 and S15 was not clear. We have no Py data for those samples because we could not analyze these samples, even after having grind them strongly (30 Hz for 3 min using a stainless steel Retsch ball mill). Indeed, these sediments are corresponding to coarse sand, and thus it was impossible to prepare a bowl of 200 $\mu$g with our capillary system which is used to weight accurately ($\pm$ 20 $\mu$g) and transfer our samples to the Py cup. Hence, we agree by clarifying this technical issue in our manuscript by replacing the sentence lines 185-186 by "Because these sediment samples are too coarse (predominantly sand) for Py-GC/MS analysis according to our method based on 200 $\pm$ 20 $\mu$g analyzed sample mass, they are excluded from the data analyses and discussion".

We could not have analyzed those two sediments with higher sediment mass in order to include the resulting data in our statistical analyses because we employed a data processing pipeline to automatically integrate the peaks and extract the corresponding mass spectra. For getting accurate and precise peak integration and extracted mass spectra from this data processing pipeline, the baseline of the Py chromatogram has to be similar between the different sample Py chromatogram. However, when injecting strongly different sample mass the baseline is strongly affected.

8. "Given the effort that has been put into analyzing the geochemical data I think it would have been better to determine the %TOC using elemental analysis as was done in Tolu et al 2015. LOI is not a very precise way to determine TOC. the pyrolysis results could be expressed per TOC."

Although it would have been great to determine TOC content, LOI is a very good indicator of OM content, which has been, and is still, widely used. Moreover, it does not make more sense to express the pyrolysis results per TOC than as relative abundance (%) based on the sum of peak areas of identified compounds, as done here and all previous studies using Py-GC/MS in environmental matrix. Indeed, in others of our studies where we look at OM composition by Py-GC/MS in long sediments records (for Holocene paleo-re construction) or varved sediments (to study OM diagenesis over 30 years), we could observe that the sums of peak areas of identified compounds are always significantly positively correlated to TOC content (Figure R1 in the supplement file). In this study on the spatial variability of OM composition in Härsvatten, there is also a significant correlation between the sum of peak areas of identified compounds and LOI (Figure R2 in the supplement file). Therefore, using the pyrolysis-GC/MS data expressed per TOC or as relative abundance will give the same trends and thus the same data interpretation and conclusions. However, it is much more convenient for a reader to compare and check the trends we discussed, in term of OM composition between a large number of compounds groups and sediment samples, when the data are presented as relative abundance than as peak areas normalized by TOC or LOI.
9. "Table 2: Give references to the information that is underlying the grouping. This table looks a bit rushed. The compound classes are not always clear and confusingly grouped. I object against the terminology used to group the compounds and the way how the grouping is organised. The authors must decide how to group the compounds and stick to a very clear well explained system of doing so. What are "Big furans"? please improve. Maleimie = Maleimide Proteins as a class of pyrolysis compounds is not possible, I suspect protein derived is meant here. Lipids is the wrong term other compounds in the list are lipids too (eg hopanoids). Aliphatics is a better label. Steroids and hopanoids have different biological origins, for a statistical approach such as what is presented here it would make a lot of sense to keep these separated. This would help with statistical analysis. A term such as lignin has a clear origin associated with it, a term such as "lipid" or "high mass compound" caries no such value. The table as given in the supplement is much more useful. It is very important to see what contributes to these categories. Without it, the methodology becomes a black box. I suggest to combine table 2 and S1. This will avoid that the reader is wonders about what might be in a "big furan" category. I would suggest to rename some of the categories and rethink the groupings. With such a small change, our PCA results are not going to change significantly."

We agree with the referee that better homogeneity in the terminology used to present the compounds groups will improve our manuscript and will be useful for the readers. We will thus rename some of the categories in the revised manuscript, and will change Table 2. However, we would like to underline that our grouping is homogeneous because it is based on similarity in the molecular structure, and that the example given by referee 2 to discuss our grouping, i.e. "Steroids and hopanoids have different biological origins, for a statistical approach such as what is presented here it would make a lot of sense to keep these separated", is not fully correct. Indeed, while we present the "steroids" and "hopanoids" within the same groups in Table 2 of the manuscript, these two groups of compounds have been kept separated for the statistical analyses (cf. Fig. 3 in the manuscript, which presents the output of the PCA analysis). The

only major change that could be done in our grouping would be to gather the "furans" with the "big-furans", this latter group being mentioned by the referee 2 for its unclear terminology. Therefore, our PCA results are not going to change.

10. "I would like to see Retention Index values for each of the listed compounds in the table. This is very important as all compounds have been identified using the NIST database (according to the text) and this can be very unreliable. RI values together with the mass spectra (as have correctly been given) will give extra confidence. The mass spectra given are only a rough summary and do not have much value for a reader but with RI values it becomes much more solid."

Adding the retention index (RI) for each of the identified peak (compounds listed in Table S1) will effectively provide extra confidence to our identification and could be useful for future users of Py-GC/MS. We will thus add in the revised supplementary information, when possible, the RI values of the identified compounds resulting from our analyses and the corresponding reference RI values, i.e. RI values determined by previous studies using similar GC operating conditions (non-polar column and temperature-ramp GC program) which are provided by the NIST in their "NIST Chemistry Webbook" website (http://webbook.nist.gov/chemistry/) and/or in the 'NIST/EPA/NIH 2011' library included in the software "NIST MS Search v.2.0" (cf. Table R1 in the supplement file). Because the RI of a certain chemical compound is its retention time normalized to the retention times of adjacently eluting n-alkanes, we unfortunately cannot provide the RI values for each of our identified compounds. Indeed, we only detected n-alkanes from the "nonane, C9:0" which elutes at 282.2 seconds. This makes impossible to calculate the RI for compounds eluting before 282.2 s, i.e. for 15 compounds belonging to carbohydrates, acetamino-sugars or N-compounds. On the contrary, there are ~50 compounds for which we have determined RI values but could not find any corresponding reference values. For the remaining ones (i.e. ~70 compounds), we obtain a very good match between our determined RI values and the reference ones (cf. Table R1 in the supplement file), showing that our peak identification is very reliable. The RI

values give for sure extra confidence, but there is a large number of Py organic compounds that can be identified but for which there are no previously reported RI values. For those, we believe that a careful comparison between the experimental mass spectra and theoretical mass spectra (available from NIST MS Search v2.0 software and 'NIST/EPA/NIH 2011' libraries) or those given in published studies is the best way to identify peaks in a reliable way. As we have discussed in our methodology paper, a compound can be proposed by the software to match the experimental spectra with a R. match value above 750, or even 800, while the unknown and proposed theoretical spectra do not obviously match according (cf. Tolu et al., 2015 ACA, 880, 93-102).

11. "There are quite a few unknowns. I would like to challenge the authors: if they are not certain about the ID of a compound, how can you put a compound in a specific category?"

First we will rectify mistakes we have made for some compounds which have been labelled as unknown (which does not mean our ID is uncertain) while the molecular structure is actually known, i.e. for levoglucosenone, levomannosan, levogalactosan, levoglucosan, Stigmasta-3,5-dien-7-one. Thereafter, there will be only 14 compounds for which the molecular structure remains unclear (over >150 identified Py compounds), i.e. anhydrohexose, 3 acetamido-4-pyrone, Oxazoline structure, diketodipyrrole, six alkylamides, and four hopanoids (Trisnorhopane, Norhopene (triterpene C29), norhopane (C30/C31?), Norhopene (C30/C31?)). The "anhydrohexose" has been assigned to a mass spectrum corresponding to a mass spectrum reported by Faix et al., 1991 that have specifically studied the pyrolytic products of woods polysaccharides. Therefore, although we do not know exactly the molecular structure of this compound, we know this compound is a pyrolytic product of carbohydrates/polysaccharides. Similarly, we assigned acetamido-4-pyrone, oxazoline structure, and diketodipyrrole based on mass spectra reported in previous studies aiming at identifying the pyrolytic nitrogen-containing products of chitin and/or proteins. For those compounds that are based on published mass spectra instead of the

NIST library, a reference is given in Table S1 of the supplementary information. The alkylamides are molecules made of C, H, N and O atoms which present highly specific mass spectra in electron impact-MS; their mass spectra is made of two m/z with high abundance (i.e. m/z 59 and 72) and many m/z of very low abundance including the m/z corresponding to the molecular ion of the specific alkylamide, such as 227 for tetradecanamide (alkylamide C14) or 255 for hexadecanamide (alkylamide C16). We did observe peaks with such mass spectra but identifying the highest m/z corresponding to the molecular ions was too uncertain because their signal intensity was too close from the background. Hence, we have only assigned these peaks to "alkylamide" rather than to a certain alkylamide compound. Similarly, the hopanes and hopenes are well known to have one or two important and highly specific m/z in their mass spectra (i.e. 191 and 177), but the m/z corresponding to the molecular ions are of very low abundance and too close from the background as well.

Please also note the supplement to this comment:
http://www.biogeosciences-discuss.net/bg-2016-361/bg-2016-361-AC2-supplement.pdf

---

## Author Response (AR1)

**Response to the Associate Editor comments after reviewing process and marked-up versions of the revised manuscript and the supplementary information**

Dear Editor,

We appreciate that you find our answer to the comments of the two reviewers thorough, and are thankful for your additional comments that helped to further improve our manuscript. We have made all the changes in the manuscript that we have indicated we would do in our answer to the reviewer's comments. We have also made additional changes with respect to your comments about these reviews. Please find, below, i) a point by point answer to your comments with a description of the corresponding modifications in the manuscript, and ii) the marked-up revised manuscript and supplementary information.

1. "In reply to some of the comments of referee one you stated that a more extensive discussion on the Fe, Mn, and Al distributions in the lake falls outside the scope of this study. The scope of this study more or less defined as determining the spatial heterogeneity in the lake indicating that taking a single sample and scaling up to the entire surface area of the lake is not the way to go. I do agree that an extensive discussion is not warranted here, but you do suggest some mechanisms that might play a role and you specify these mechanisms. This makes the argument of the scope of the study invalid, I think, if something falls outside the scope of the work you could argue it should not be mentioned at all. It is not a very good excuse. Therefore I would like to ask you to either make the effort of writing a more extensive discussion on different possible mechanisms or you make it very clear that the mechanisms mentioned are just of few possibilities and that there might be others such as ... (see review and reply)."

We have now added a short discussion of the other possible factors and processes that could be involved in the spatial distribution of the redox elements (Fe, Mn) in the sub-section 3.3.2 (cf. lines 642-646 in the marked revised manuscript).

We have also changed the titles of the sub-section 3.3 to emphasize that the discussion of our data is mainly focused on the spatial distribution of the molecular composition of the sediment OM. The title "3.3 Combined spatial patterns of elemental and organic biogeochemistry" has been replaced by "3.3 Factors and processes involved in the spatial distribution of OM molecular composition" (cf. line 521 in the marked revised manuscript). The title "3.3.1 Spatial variability in the sediment composition in the main south basin" has been replaced by "3.3.1 Spatial variability in the main south basin" (cf. line 541 in the marked revised manuscript).

2. "The other thing I noticed is that the "whole-lake" concept creates some response from both reviewers in different ways. As for the title I agree with the suggested changes. Throughout the manuscript, just realize that you did not analyze the whole lake. Rather than just the typical one sediment sample you analyzed more samples from different sites to get an idea of the heterogeneity.

**This is very interesting, and provides a much better picture than analyzing just one sample, but you did not analyze the whole lake or even the all the lake sediments."**

We agree and have removed the term "whole-lake" everywhere in the revised manuscript (e.g., line 63 in the marked revised manuscript), except when we compare and discuss the average values of the different clusters with the average values of all analyzed sediment samples. We kept the term "whole lake average" for saying "averages of all analyzed sediment samples" because more convenient to read. However, we have clearly defined this term the first time we employ it in the revised manuscript (cf. lines 319-321 in the marked revised manuscript), as well as in Table 3 and in Tables S3 and S5 in the revised supplementary information.

**3.** "One question I had is what about micro-heterogeneity did you analyze multiple "replicates" from the same sample to see how large the variation is within one sample? 200µg is a relatively small amount."

 $200 \ \mu g$  is a very small amount, but all sediment samples were first very finely ground and could be easily mixed and weighted for Py-GC/MS analyses, except the sediments at sites M5 and S14, which were too coarse (for which we could not get Py-GC/MS data).

We did not measure replicates for the sediment samples in this study, but we have tested extensively the reproducibility for the different identified organic compounds when using only 200  $\mu$ g of sediment in our methodological paper published in Analytica Chimica Acta. The reproducibility (i.e. relative standard deviation from triplicate analyses) we measured for the 233 identified compounds (including those that have been identified in Härsvatten sediment samples) was 5.5 ± 4.3%, with 90% of the RSD values within 10% and 98% within 15%.

**4.** "You suggest that you looked at the whole matrix rather than only specific compounds or classes. And although I agree that pyrolysis might be one of the better tools to do this, the method still has biases and issues like all other analytical methods. If it doesn't fit in your analytical window you don't see it. The discussion about the pyrolysis temperature makes that quite clear, different temperatures show (slightly) different results, so please be careful in how you phrase what your pyrolysis results reflect. That brings me to a comment from referee two about other studies on OM composition studies. Other groups have investigated complex organic matter from lake and other sediments through a whole range of different techniques including pyrolysis (with and without TMAH), lipid extractions, GC and LC mass spectrometry etc etc. I think your approach of analyzing every samples the same way works fine for this manuscript, but I think it is good to put your own work in a larger, historical perspective by shortly mentioning other studies. Pyrolysis has been around for quite some time and it is good that this would be reflected in your reference list."

We agree that saying we look at the whole OM matrix was not accurate enough. Therefore, we replaced the term "whole OM matrix" that was used in the introduction only. Please, see lines 66-68 and 127-128 in the marked revised manuscript.

We have also added an important paragraph in the introduction of the revised manuscript where the existing methods to characterize OM molecular composition are discussed in order to put our work and data provided by Py-GC/MS in a larger and historical perspectives (cf. lines 69-114 in the marked revised manuscript).

Moreover, few sentences in the material and method section have been added to explain that Py has been used previously for different environmental samples types and with different pyrolysis temperatures than the one we used in this study (cf. lines 207-226 in the marked revised manuscript). Thus, our reference list better reflects the literature (11 added references for Py-GC/MS applied to environmental samples).

Finally, we believe we have been extremely careful in interpreting and phrasing what our pyrolysis results reflect. Given the statistical approach we use and our experience in using Py-GC/MS, we think we have not over-interpreted our data in this manuscript. Notably, we are not concluding by claiming that our study provides how the molecular composition of OM is in the different areas of lake basin due to specific processes, but our conclusion is that OM molecular composition can vary significantly within a single lake system in relation to factors and processes that are common to lake ecosystems, and our main message for future research is that given these results, biogeochemical processes known to be influenced by OM composition such as C mineralization, Hg methylation and P desorption, should not only be studied by comparing between-lake variability using sediment of the deeper area as generally done, but the in-lake spatial variability should also be investigated.

**5.** "As referee two mentions, be very careful with steroids and hopanoids, they come very different sources. Looks can be deceiving. Be very careful with assigning different classes and binning compounds together and using only or mainly the NIST database for compound identifications. Mistakes are easily made."

We always seek to be very careful in i) assigning compounds to mass spectra from the NIST database; ii) classifying the different compounds into OM groups; and iii) discussing the origin of the different compounds; and we are very much aware about the mistakes that can be done when doing all these steps. To get accurate peak identification, organic compounds classification and interpretation about their origin and degradation status, we have carefully researched a large number of articles, i.e.,

i) almost all papers where Py-GC/MS was employed, and at least all papers on Py-GC/MS applied to environmental samples, i.e. soils, peat, plants and sediments for reaching different objectives, e.g. paleoreconstruction, variability in OM composition and identification of Py products from standards or specific OM fractions.

ii) a wide numbers of papers using other techniques and mainly mass spectrometry, such as specific extractions and LC-MS or GC-MS analyses for determination of specific biomarkers from the

different OM biochemical classes or Orbitrap-MS and FT-ICR-MS for the characterization of the whole OM matrix in liquid environmental samples.

This, we hope, is better reflect in the revised manuscript with the new paragraph in the introduction (cf. lines 69-115 in the marked revised manuscript and answer to comment 4 above), in the material and method (cf. lines 206-224 and text below) and the associated additional 25 references.

Our list of identified compounds is highly similar to the lists of identified Py-GC/MS organic compounds in soil, peat and sediment samples reported in published papers from other research groups, regarding both the organic compounds that are present and how these compounds are grouped into different OM families (i.e. carbohydrates, N-compounds, *n*-alkenes, chlorophyll, steroids, hopanoids...). This information has been added in the material and method section (cf. lines 206-224 in the marked revised manuscript). Our compound grouping into 41 groups is less common; and we have done this grouping for the sake of making the presentation of the data and the associated discussion more constrained and to avoid over-interpreting individual compounds. Given the comment 9 of the reviewer 2, we have now explained why and how the groupings of the 160 identified compounds into 41 groups have been done (cf. lines 344-367).

Regarding the specific comment on the steroids and hopanoids, we agree we have made a mistake by gathering them in Table 2 of the manuscript (this mistake has been now corrected; cf. Table 2 in the revised manuscript). But, we would like to point out that these compounds were neither grouped for the statistical analysis or in the detailed lists of identified organic compounds (Table S1 in SI). Moreover, in the text, we are clearly saying that the Py products of steroids in our samples mainly originate from algal production while hopanoids are well known to be of prokaryotes, especially bacteria, origin. Please see in the marked revised manuscript, lines 437-439 "*
[revised manuscript text omitted]
., 2001). As an analytical tool to characterize OM composition, Py-120 GC/MS is a compromise between the quantitative, molecular information obtained from wet 121 chemical extractions associated with liquid chromatography (LC)-MS or GC-MS analyses, and the 122 qualitative, non-molecular information provided by high-throughput techniques such as visible-123 near-infrared spectroscopy or 'RockEval' pyrolysis. Beyond the rapidity in terms of analysis and 124 data treatment, our Py GC/MS method yields semi-quantitative data on >100 organic compounds belonging to different biochemical classes (e.g., lignins, lipids, chlorophylls, carbohydrates, 125 126 Nitrogen (N)-compounds), which makes it possible to explore the overall molecular composition of 127 OM (Tolu 
[revised manuscript text omitted]
., 2010). as well as acetamido-451 sugars, which derive from Py products of chitin of micro-organisms exoskeletons (Gupta et al., 452 2007), and hopanoids that are of prokaryotes, mainly bacteria, origin (Meredith et al., 2008; 453 Sessions et al., 2013) also have positive loadings on PC4OM, while no compounds are significantly 454 negatively correlated to PC4OM (Fig. 3b). Therefore, PC4OM reflects OM input from in-lake algae 455 and micro-organisms (e.g., zooplankton). Steroids, which have not yet been reported by Py-GC/MS 456 in aquatic matrices, have positive loadings on this PC4OM suggesting that the steroids released by 457 Py in aquatic samples are mainly of algal origin.

[revised manuscript text omitted]

541 3.3.1 Spatial variability in the sediment composition in the main south basin

542

543 As shown previously for OM (as % LOI) and Pb (Bindler et al. 2001), there is a physical and 544 inorganic geochemical gradient from shallower to deeper waters reflecting sediment focusing in the south basin of Härsvatten. B.D. and bSi decrease from shallower (clustergeo6) to intermediate 545 (clustergeo2) to deeper areas (clustergeo5), whereas there is a progressive enrichment in organic 546 547 matter and trace elements with increasing water depth (Fig. 1c; Table 1). For example, B.D. decreases from ~0.07 to 0.03 g cm-3 while OM and Hg increase from ~34 to 52 % and from ~230 to 548 920 ng g-1, respectively, in shallower versus the deepest locations. At intermediate depths 549 (clustergeo2), OM, B.D., bSi and most trace metals (i.e., Cu, Ni, Hg, Zn) are between those of 550 shallow and deep locations. Sediment focusing is thus an important process for sediment 551 geochemistry in the large, deep basin of Härsvatten, which presents a relatively simple 552 morphometry. The sediments found at shallower (<11 m; clustergeo 6), intermediate (11-21 m; 553 clustergeo 4) and deeper water depths (>23 m; clustergeo 5) would correspond approximatively to 554 555 erosion, transportation and accumulation bottoms, respectively (Håkanson, 1977). The bSi decline, from ~15 to 4 %, would reflect indicates a decrease of diatom production with depth due to 556 557 increasing light attenuation, and thus suggests the fact that the diatom assemblage is dominated by 558 benthic diatoms, such as in shown for many acidified lakes, such as the surrounding lakes in the 559 Svartedalen nature reserve, has been shown to be dominated almost exclusively by benthic diatoms, 560 with a near absence of planktonic diatoms (
[revised manuscript text omitted]

- Tolu, J., Gerber, L., Boily, J. F. and Bindler, R.: High-throughput characterization of
  sediment organic matter by pyrolysis-gas chromatography/mass spectrometry and multivariate
  curve resolution: A promising analytical tool in (paleo) limnology, Anal. Chim. Acta, 880, 93-102,
- 942 2015.

943 Tranvik, L. J., Downing, J. A., Cotner, J. B., Loiselle, S. A., Striegl, R. G., Ballatore, T. J.,

944 Dillon, P., Finlay, K., Fortino, K., Knoll, L. B., Kortelainen, P. L., Kutser, T., Larsen, S., Laurion,

945 I., Leech, D. M., Leigh Mccallister, S., Mcknight, D. M., Melack, J. M., Overholt, E., Porter, J. A.,

946 Prairie, Y., Renwick, W. H., Roland, F., Sherman, B. S., Schindler, D. W., Sobek, S., Tremblay, A.,

947 Vanni, M. J., Verschoor, A. M., Von Wachenfeldt, E. and Weyhenmeyer, G. A.: Lakes and

reservoirs as regulators of carbon cycling and climate, Limnol. Oceanogr., 54, 2298-2314, 2009.

Trolle, D., Zhu, G. W., Hamilton, D., Luo, L. C., Mcbride, C. and Zhang, L.: The influence of
water quality and sediment geochemistry on the horizontal and vertical distribution of phosphorus
and nitrogen in sediments of a large, shallow lake, Hydrobiologia, 627, 31-44, 2009.

Valdes, F., Catala, L., Hernandez, M. R., Garcia-Quesada, J. C. and Marcilla, A.:
Thermogravimetry and Py-GC/MS techniques as fast qualitative methods for comparing the
biochemical composition of Nannochloropsis oculata samples obtained under different culture
conditions, Bioresource Technol., 131, 86-93, 2013.

956 Vancampenhout, K., Wouters, K., Caus, A., Buurman, P., Swennen, R. and Deckers, J.:957 Fingerprinting of soil organic matter as a proxy for assessing climate and vegetation changes in last

958 interglacial palaeosols (Veldwezelt, Belgium), Quaternary Research, 69, 145-162, 2008.

Vogel, H., Wessels, M., Albrecht, C., Stich, H. B. and Wagner, B.: Spatial variability of
recent sedimentation in Lake Ohrid (Albania/Macedonia), Biogeosciences, 7, 3333-3342, 2010.

Wagner, S., Jaffé, R., Cawley, K., Dittmar, T. and Stubbins, A.: Associations between the
molecular and optical properties of dissolved organic matter in the Florida Everglades, a model
coastal wetland system, Frontiers in Chemistry, 3, 2015.

Wakeham, S. G., Lee, C., Hedges, J. I., Hernes, P. J. and Peterson, M. L.: Molecular
indicators of diagenetic status in marine organic matter, Geochim. Cosmochim. Ac., 61, 5363-5369,
1997.

967 Yin, H., Feng, X. H., Qiu, G. H., Tan, W. F. and Liu, F.: Characterization of Co-doped
968 birnessites and application for removal of lead and arsenite, J. Hazard. Mater., 188, 341-349, 2011.

- 969 Zheng, Y. H., Zhou, W. J. and Meyers, P. A.: Proxy value of n-alkan-2-ones in the Hongyuan
- 970 peat sequence to reconstruct Holocene climate changes on the eastern margin of the Tibetan
- 971 Plateau, Chem. Geol., 288, 97-104, 2011.
- 972 Zhu, M. Y., Zhu, G. W., Li, W., Zhang, Y. L., Zhao, L. L. and Gu, Z.: Estimation of the algal-
- 973 available phosphorus pool in sediments of a large, shallow eutrophic lake (Taihu, China) using
- 974 profiled SMT fractional analysis, Environ. Pollut., 173, 216-223, 2013.

975

976

**Tables**

| 1 | 0  |  |
|---|----|--|
|   | Ο. |  |
|   | -  |  |

|                            | , , , , , , , , , , , , , , , , , , ,  | Whole sample c          | ollecti             | on except t  | he two out       | liers                              | 0                                                 | Outliers (      | (M4, S15)             |                 |
|----------------------------|-----------------------------------------------|-------------------------|---------------------|--------------|------------------|------------------------------------|---------------------------------------------------|-----------------|-----------------------|-----------------|
|                            | Unit                                          | $Av.^{a} \pm sd^{b}$    | CV c     | Median       | A:M d | Min e -Max f | $Av. \pm sd$                                      | CV c | Median                | Min-Max         |
| W.D.                       | m                                             | 9 ± 7            | 74                  | 7            | 1.23             | 2-25                               | 3.4 ± 0.6                                  | 19              | 3                     | 2.9-3.8         |
| B.D.                       | g cm -3                            | $0.06 \pm 0.02$         | 38                  | 0.06         | 1.05             | 0.02-0.13                          | 0.67 ± 0.09                                | 14              | 0.06                  | 0.61-0.74       |
| bSi                        | %                                             | 13 ± 6           | 48                  | 12           | 1.05             | 4-25                               | $1.9 \pm 0.2$                                     | 0               | 12                    | 1.7-2.0         |
| LOI                        | %                                             | 38 ± 10          | 26                  | 37           | 1.01             | 10-58                              | $3.6 \pm 0.8$                                     | 20              | 37                    | 3.0-4.1         |
| [S]                        | mg kg -1                           | 11876 ± 5920     | 50                  | 11305        | 1.05             | 4685-29190                         | 2570 ± 552                                 | 21              | 10610                 | 2180-2960       |
| [Br]                       | mg kg -1                           | $149\pm35$              | 23                  | 152          | 0.99             | 71-225                             | 16 ± 7                                     | 44              | 148                   | 11-21           |
| [Cu]                       | mg kg -1                           | 34 ± 13          | 37                  | 32           | 1.07             | 12-75                              | 9 ± 3                                      | 31              | 31                    | 7-11            |
| [Ni]                       | mg kg -1                           | $19 \pm 4$              | 24                  | 19           | 0.99             | 10-27                              | $12 \pm 4$                                        | 35              | 19                    | 9-15            |
| [Hg]                       | µg kg⁻¹                                       | 337 ± 202        | 60                  | 286          | 1.18             | 117-1152                           | 28 ± 9                                     | 33              | 274                   | 21-34           |
| [Pb]                       | mg kg -1                           | $192 \pm 74$            | 39                  | 184          | 1.05             | 58-422                             | 22 ± 16                                    | 76              | 178                   | 10-33           |
| [Zn]                       | mg kg -1                           | 219 ± 108        | 49                  | 207          | 1.06             | 43-445                             | 50 ± 16                                    | 31              | 200                   | 39-61           |
| [Al]                       | %                                             | $3 \pm 1$               | 17                  | 3            | 1.06             | 2-4                                | $\textbf{5.67} \pm 0.01$                          | 0.1             | 3                     | 5.66-5.67       |
| [Y]                        | mg kg -1                           | $25\pm 8$               | 32                  | 25           | 1.01             | 7-43                               | $20 \pm 4$                                        | 18              | 25                    | 17-22           |
| [Fe]                       | %                                             | 5 ± 3            | 65                  | 4            | 1.26             | 1-12                               | $3.4 \pm 0.1$                                     | 4               | 4                     | 3.3-3.5         |
| [As]                       | mg kg -1                           | $35 \pm 20$             | 56                  | 28           | 1.26             | 5-73                               | <dl< td=""><td></td><td>27</td><td>0-0</td></dl<> |                 | 27                    | 0-0             |
| [ P ]               | mg kg -1                           | 1624 ± 741       | 46                  | 1401         | 1.16             | 655-3769                           | 949 ± 57                                   | 6               | 1389                  | 908-989         |
| [Mn]                       | mg kg -1                           | 729 ± 1690       | 232                 | 180          | 4.06             | 94-7981                            | $\textbf{1060} \pm 845$                           | 80              | 184                   | 462-1657        |
| [Co]                       | mg kg -1                           | 19 ± 15          | 77                  | 14           | 1.39             | 5-76                               | 17 ± 9                                     | 56              | 14                    | 10-23           |
| [Ca]                       | mg kg -1                           | 5261 ± 1306      | 25                  | 5213         | 1.01             | 2860-9300                          | $\textbf{26540} \pm 7566$                         | 29              | 5283                  | 21190-31890     |
| [K]                        | mg kg -1                           | 4426 ± 1020      | 23                  | 4485         | 0.99             | 2420-6140                          | $10510 \pm 2616$                                  | 25              | 4580                  | 8660-12360      |
| [Mg]                       | mg kg -1                           | $\textbf{1488} \pm 354$ | 24                  | 1500         | 0.99             | 870-2130                           | 7495 ± 3599                                | 48              | 1515                  | 4950-10040      |
| [Na]                       | mg kg⁻¹                                       | 1795 ± 659       | 37                  | 1743         | 1.03             | 440-3380                           | $\textbf{10695} \pm 587$                          | 5               | 1783                  | 10280-11110     |
| [Si inorganic ] | %                                             | $11 \pm 4$              | 33                  | 11           | 1.06             | 4-21                               | 23 ± 1                                     | 3               | 11                    | 22-23           |
| [Sr]                       | mg kg -1                           | 55 ± 16          | 29                  | 55           | 1.01             | 27-116                             | 235 ± 24                                   | 10              | 55                    | 218-252         |
| [Ti]                       | mg kg -1                           | $2115 \pm 495$          | 23                  | 2200         | 0.96             | 997-2870                           | 4357 ± 2348                                | 54              | 2215                  | 2697-6017       |
| [V]                        | mg kg -1                           | 63 ± 15          | 23                  | 60           | 1.05             | 36-101                             | 75 ± 23                                    | 31              | 60                    | 58-91           |
| [Zr]                       | mg kg -1                           | $101 \pm 31$            | 31                  | 100          | 1.01             | 39-160                             | $158 \pm 6$                                       | 4               | 103                   | 153-162         |
| a Av.: avera    | ige; b sd: star                    | ndard deviation;        | c CV: c  | oefficient o | of variation     | calculated as re                   | lative standard de                                | viation in      | n %; d A:M | : ratio between |
| average and                | l median Mi                                   | n.: minimal valu        | es; e Ma | ax.: maxima  | al value         |                                    |                                                   |                 |                       |                 |

Table 1. Summary statistics for sediment elemental geochemistry

| Telutive ubuildui                            |                             | $Av^{a} \pm sd^{b}$  | <del>CV</del> e | Median         | A:M d | Min e -Max e |
|----------------------------------------------|-----------------------------|----------------------|----------------------------|----------------|------------------|------------------------------------|
|                                              | Furans                      | <del>15 ±</del> 4    | <del>30</del>              | 14             | <del>1.06</del>  | 988
8-28                        |
|                                              | Big_furans           | 4.1 ± 1.2     | <del>29</del>              | <del>4.0</del> | <del>1.03</del>  | <del>0.8 76520</del>        |
| Carbobydrates                                | Pyrans                      | 3.4 ± 1       | <del>30</del>              | <del>3.2</del> | <del>1.06</del>  | <del>1.2 5.3</del>                 |
| Carbonyarates                                | Dianhydrorhamnose    | 1.6 ± 0.5     | <del>28</del>              | <del>1.7</del> | <del>0.99</del>  | <del>0.3 297</del> 90              |
|                                              | Levoglucosenone      | 2.2 ± 0.4     | <del>20</del>              | <del>2.2</del> | <del>1.00</del>  | <del>1.3-3.1</del>                 |
|                                              | Levosugars           | <del>3.7 ± 2.6</del> | 71                         | <del>2.5</del> | <del>1.46</del>  | <del>0.8-19</del> 91               |
| Chitin                                       | Acetamidosugars      | <del>2.5 ± 1</del>   | <del>40</del>              | <del>2.6</del> | <del>0.98</del>  | <del>0.2</del> -4.2                |
|                                              | <del>(alkyl)pyridines</del> | 0.3 ± 0.1     | <del>34</del>              | <del>0.3</del> | <del>0.95</del>  | <del>0.1 0.9</del> 2               |
|                                              | Pyridines_0                 | 0.7 ± 0.1     | <del>18</del>              | <del>0.7</del> | <del>1.00</del>  | 0.2 0.9                            |
|                                              | <del>(alkyl)pyrroles</del>  | $2.4 \pm 0.5$        | 22                         | <del>2.4</del> | <del>1.01</del>  | 993
<del>1.7-3.5</del>          |
|                                              | Pyrroles_0                  | 1.0 ± 0.2     | <del>25</del>              | <del>0.9</del> | <del>1.04</del>  | <del>0.5 1.4</del>
001          |
| <del>N-compounds</del>                       | Maleimie & succinimide      | $1.2 \pm 0.3$        | <del>29</del>              | <del>1.2</del> | <del>0.98</del>  | <del>0.2-1.7</del>                 |
|                                              | Aromatic N                  | 0.8 ± 0.3     | <del>36</del>              | <del>0.8</del> | <del>1.03</del>  | <del>0.3 lop</del> 5               |
|                                              | Indoles                     | 1.5 ± 0.4     | <del>24</del>              | <del>1.5</del> | <del>1.03</del>  | 0.5 3.1                            |
|                                              | Diketodipyrrole      | 0.8 ± 0.2     | 22                         | <del>0.8</del> | <del>1.01</del>  | <del>0.4 19</del> 96               |
|                                              | Proteins                    | 1.5 ± 0.4     | <del>30</del>              | <del>1.5</del> | <del>1.02</del>  | <del>0.3-2.6</del>                 |
|                                              | Alkylamides          | 0.6 ± 0.3     | <del>51</del>              | <del>0.6</del> | <del>1.06</del>  | <del>0.1–199</del> 7               |
| <del>Phenols and</del>
<del>Lignins</del> | Phenols              | <mark>8 ± 1</mark>   | <del>15</del>              | 8              | <del>1.02</del>  | 4.4-11.4                           |
|                                              | Syringols                   | 0.5 ± 0.4     | <del>83</del>              | <del>0.4</del> | <del>1.32</del>  | <del>0.1–1.9</del>                 |
|                                              | Guaiacols            | 3.6 ± 2.3     | <del>65</del>              | <del>2.9</del> | <del>1.2</del> 4 | <del>1.1-13.5</del>                |
| Chlorophylls                                 | Pristenes            | 2.7 ± 0.8     | <del>28</del>              | <del>2.8</del> | <del>0.97</del>  | <del>0.4-4.6</del>                 |
| Chlorophyns                                  | Phytadienes          | 1.9 ± 0.7     | <del>35</del>              | <del>1.8</del> | <del>1.04</del>  | <del>0.2-3.6</del>                 |
|                                              | <del>C9-16:1</del>          | 3.5 ± 0.8     | 23                         | <del>3.6</del> | <del>0.98</del>  | <del>1.8 5.1</del>                 |
|                                              | <del>C17-C22:1</del>        | <del>6 ± 1</del>     | <del>17</del>              | <del>6.2</del> | <del>0.97</del>  | <del>3.5-8.9</del>                 |
|                                              | <del>C23-26_1</del>         | 2.9 ± 0.9     | <del>32</del>              | 2.7            | <del>1.09</del>  | <del>0.6-5.4</del>                 |
|                                              | <del>C27-28:1</del>         | 0.8 ± 0.4     | 47                         | <del>0.7</del> | <del>1.10</del>  | <del>0.1–1.4</del>                 |
|                                              | <del>C13-16:0</del>         | $2.5 \pm 0.6$        | <del>23</del>              | 2.5            | <del>1.03</del>  | <del>1.3</del> -4.1                |
| Lipids                                       | <del>C17-22:0</del>         | 3.9 ± 0.8     | <del>21</del>              | <del>4.0</del> | <del>0.98</del>  | <del>1.6-5.4</del>                 |
|                                              | <del>C23-26:0</del>         | 2.8 ± 1.4     | <del>49</del>              | <del>2.7</del> | <del>1.07</del>  | <del>1.4-8.8</del>                 |
|                                              | <del>C27-35:0</del>         | 4.3 ± 3.5     | <del>80</del>              | <del>3.6</del> | <del>1.20</del>  | <del>1.1-21.3</del>                |
|                                              | <del>2K C13-17</del>        | 1.3 ± 0.4     | <del>33</del>              | <del>1.4</del> | <del>0.96</del>  | <del>0.6-2.2</del>                 |
|                                              | <del>2K C19-22</del>        | 0.3 ± 0.1     | <del>45</del>              | <del>0.3</del> | <del>0.97</del>  | <del>0-0.8</del>                   |
|                                              | <del>2K C23-31</del>        | 1.3 ± 0.8     | <del>62</del>              | <del>1.1</del> | <del>1.24</del>  | <del>0.1 3.3</del>                 |
| High molecular                               | Steroids             | 1.2 ± 0.9     | <del>70</del>              | <del>1.1</del> | <del>1.10</del>  | <del>0-4.3</del>                   |
| mass compounds                               | Tocopherols          | 0.3 ± 0.3     | <del>106</del>             | <del>0.2</del> | <del>1.75</del>  | <del>0-1.5</del>                   |
|                                              | Hopanoids                   | 1.3 ± 0.4     | <del>31</del>              | <del>1.4</del> | <del>0.94</del>  | <del>0.2-1.9</del>                 |
|                                              | Benzene                     | 0.9 ± 0.4     | <del>43</del>              | <del>0.8</del> | 1.14             | <del>0.4-2.5</del>                 |
|                                              | Benzaldehyde         | 0.6 ± 0.3     | 41                         | <del>0.6</del> | <del>1.08</del>  | <del>0.3-1.5</del>                 |
| <del>(poly)aromatics</del>                   | Acetylbenzene        | 1.1 ± 0.4     | <del>39</del>              | <del>1.0</del> | <del>1.10</del>  | <del>0.6-2.3</del>                 |
|                                              | Alkylbenzenes C3-9          | 1.9 ± 0.5     | <del>23</del>              | <del>1.8</del> | <del>1.07</del>  | <del>1.4-3.5</del>                 |
|                                              | Polyaromatics        | $1.4 \pm 0.4$        | <del>27</del>              | <del>1.3</del> | <del>1.04</del>  | <del>0.8-2.1</del>                 |

**Table 2.** Summary statistics for the molecular composition of sediment OM (expressed as relative abundance, %)

a-Av.: average; bsd: standard deviation; eCV: coefficient of variation calculated as relative standard deviation in %; dA:M: ratio between average and median Min.: minimal values; eMax.: maximal value

|                                        | Compounds included                                                                                                                                                             | $Av^a \pm sd$          | CV | Median | A:M  | Min-Max  |
|----------------------------------------|--------------------------------------------------------------------------------------------------------------------------------------------------------------------------------|------------------------|----|--------|------|----------|
| Carbohydrates                          |                                                                                                                                                                                |                        |    |        |      |          |
| (Alkyl)-furans &furanones              | 3-furaldehyde, 2-furaldehyde, 2-acetyl-furan, Methyl-3-furaldehyde, 2(5H)-furanone, Methyl-2-furaldehyde, Dihydro-methyl-furanone, Methyl-2(5H)-furanone, Methyl-2-furaldehyde | 15 ± 4          | 30 | 14     | 1.06 | 8-28     |
| Hydroxy- or carboxy-furans & furanones | 2-Furancarboxylic acid, methyl ester; 2,5-Dimethyl-4-hydroxy-3(2H)-furanone; 5-(hydroxymethyl)-2-furaldehyde                                                                   | 4.1 ± 1.2       | 29 | 4.0    | 1.03 | 0.8-7.5  |
| Pyrans                                 | 5,6-dihydro-pyran-2-one, 4-hydroxy-5,6-dihydro-pyran-2-one                                                                                                                     | 3.4 ± 1         | 30 | 3.2    | 1.06 | 1.2-5.3  |
| Dianhydrorhamnose                      | Dianhydrorhamnose                                                                                                                                                              | $\textbf{1.6} \pm 0.5$ | 28 | 1.7    | 0.99 | 0.3-2.7  |
| Levoglucosenone                        | Levoglucosenone                                                                                                                                                                | $2.2 \pm 0.4$          | 20 | 2.2    | 1.00 | 1.3-3.1  |
| Anhydrosugars                          | Anhydrohexose, Levogalactosan, Levoganosan, Levoglucosan                                                                                                                       | 3.7 ± 2.6       | 71 | 2.5    | 1.46 | 0.8-11   |
| Chitin derived compounds               |                                                                                                                                                                                |                        |    |        |      |          |
| Chitin-derived compounds               | Acetamide, 3-acetamido-furan, 3-acetamido-4-pyrone, Oxazoline                                                                                                                  | 2.5 ± 1         | 40 | 2.6    | 0.98 | 0.2-4.2  |
| N-compounds                            |                                                                                                                                                                                |                        |    |        |      |          |
| (Alkyl)pyridines                       | Pyridine, 2-methyl-pyridine, 3/4-methyl-pyridine                                                                                                                               | 0.3 ± 0.1       | 34 | 0.3    | 0.95 | 0.1-0.5  |
| Pyridines_O, i.e. pyridines            |                                                                                                                                                                                |                        |    |        |      |          |
| with side chain containing a           | 2-acetylpyridine, 3-acetylpyridine, 2-Methyl-5-acetoxypyridine                                                                                                                 | $\textbf{0.7} \pm 0.1$ | 18 | 0.7    | 1.00 | 0.2-0.9  |
| "C=O" function                         |                                                                                                                                                                                |                        |    |        |      |          |
| (Alkyl)pyrroles                        | Pyrrole, Methyl-pyrrole                                                                                                                                                        | $2.4 \pm 0.5$          | 22 | 2.4    | 1.01 | 1.7-3.5  |
| Pyrroles_O, i.e. pyrroles with         |                                                                                                                                                                                |                        |    |        |      |          |
| side chain containing a                | 2-formyl-pyrrole, 2-acetyl-pyrrole, 2-formyl-1-methylpyrrole                                                                                                                   | $\textbf{1.0} \pm 0.2$ | 25 | 0.9    | 1.04 | 0.5-1.4  |
| "C=O" function                         |                                                                                                                                                                                |                        |    |        |      |          |
| Pyrroledione & pyrrolidinedione        | 2,5-pyrroledione, 2,5-pyrrolidinedione                                                                                                                                         | $1.2 \pm 0.3$          | 29 | 1.2    | 0.98 | 0.2-1.7  |
| Aromatic N- compounds                  | Benzeneacetonitrile, Benzenepropanenitrile                                                                                                                                     | 0.8 ± 0.3       | 36 | 0.8    | 1.03 | 0.3-1.4  |
| Indoles                                | Indole, Methyl-indole                                                                                                                                                          | $\textbf{1.5} \pm 0.4$ | 24 | 1.5    | 1.03 | 0.5-3.1  |
| Diketodipyrrole                        | Diketodipyrrole                                                                                                                                                                | $\textbf{0.8} \pm 0.2$ | 22 | 0.8    | 1.01 | 0.4-1.2  |
| Diketopiperazines                      | Pro-Ala, Pro-Val, Pro-Val, Cyclo-Leu-Pro, Pro-Pro, Pro-Phe                                                                                                                     | $\textbf{1.5} \pm 0.4$ | 30 | 1.5    | 1.02 | 0.3-2.6  |
| Alkylamides                            | 6 alkylamides                                                                                                                                                                  | $\textbf{0.6} \pm 0.3$ | 51 | 0.6    | 1.06 | 0.1-1.7  |
| Phenols                                |                                                                                                                                                                                |                        |    |        |      |          |
| Phenols                                | Phenol, 2-methyl-phenol, 3/4- methyl-phenol, dimethyl-phenol, Ethyl-phenol, Propenyl-phenol                                                                                    | 8 ± 1           | 15 | 8      | 1.02 | 4.4-11.4 |
| Lignins                                |                                                                                                                                                                                |                        |    |        |      |          |
| Syringols                              | Syringol, 4-vinyl-syringol, 4-formyl-syringol, 4-allenesyringol, Acetosyringone                                                                                                | 0.5 ± 0.4       | 83 | 0.4    | 1.32 | 0.1-1.9  |
| Guaiacols                              | Guaiacol, Ethyl-guaiacol, 4-vinyl-guaiacol, 4-propenyl-guaiacol, Vanillin, 4-alleneguaiacol, Acetovanillone, Vanillic acid, methyl ester, Guaiacylacetone                      | 3.6 ± 2.3       | 65 | 2.9    | 1.24 | 1.1-13.5 |

**Table 2.** Summary statistics for the molecular composition of sediment OM given as relative abundances (expressed in %) of the 41 groups of Py organic compounds, which belong to 13 classes of OM that are indicated by the grey shading *(to be continued)*

| Chlorophylls       |                                                                                                                                            |                        |     |     |      |          |
|--------------------|--------------------------------------------------------------------------------------------------------------------------------------------|------------------------|-----|-----|------|----------|
| Pristenes          | Prist-1-ene, Prist-2-ene                                                                                                                   | $\textbf{2.7} \pm 0.8$ | 28  | 2.8 | 0.97 | 0.4-4.6  |
| Phytadienes        | Phytadiene 1, Phytadiene 2                                                                                                                 | $\pmb{1.9} \pm 0.7$    | 35  | 1.8 | 1.04 | 0.2-3.6  |
| n -alkenes  |                                                                                                                                            |                        |     |     |      |          |
| C9-16:1            | n -alkenes C9, C13, C14, C16                                                                                                        | $\textbf{3.5}\pm0.8$   | 23  | 3.6 | 0.98 | 1.8-5.1  |
| C17-C22:1          | n -alkenes C17, C18, C19, C20, C21, C22                                                                                             | 6 ± 1           | 17  | 6.2 | 0.97 | 3.5-8.9  |
| C23-26_1           | n -alkenes C23, C24, C25, C26                                                                                                       | $\textbf{2.9} \pm 0.9$ | 32  | 2.7 | 1.09 | 0.6-5.4  |
| C27-28:1           | n -alkenes C27, C28                                                                                                                 | $\textbf{0.8} \pm 0.4$ | 47  | 0.7 | 1.10 | 0.1-1.4  |
| n -alkanes  |                                                                                                                                            |                        |     |     |      |          |
| C10-16:0           | n -alkanes C10, C11, C12, C13, C14, C15, C16                                                                                        | $2.5 \pm 0.6$          | 23  | 2.5 | 1.03 | 1.3-4.1  |
| C17-22:0           | n -alkanes C17, C18, C19, C20, C21, C22                                                                                             | $\textbf{3.9}\pm0.8$   | 21  | 4.0 | 0.98 | 1.6-5.4  |
| C23-26:0           | n -alkanes C23, C24, C25, C26                                                                                                       | 2.8 ± 1.4       | 49  | 2.7 | 1.07 | 1.4-8.8  |
| C27-35:0           | n -alkanes C27, C28, C29, C30, C31, C32, C33, C35                                                                                   | 4.3 ± 3.5       | 80  | 3.6 | 1.20 | 1.1-21.3 |
| Alkan-2-ones       |                                                                                                                                            |                        |     |     |      |          |
| 2K C13-17          | Alkan-2-ones C13, 16, 17                                                                                                                   | $\textbf{1.3}\pm0.4$   | 33  | 1.4 | 0.96 | 0.6-2.2  |
| 2K C19-21          | Alkan-2-ones C19, 20, 21                                                                                                                   | 0.3 ± 0.1       | 45  | 0.3 | 0.97 | 0-0.8    |
| 2K C23-31          | Alkan-2-ones C23, 14, 25, 26, 27, 28, 29, 31                                                                                               | $\pmb{1.3} \pm 0.8$    | 62  | 1.1 | 1.24 | 0.1-3.3  |
| Steroids           |                                                                                                                                            |                        |     |     |      |          |
| Steroids           | Cholest-2-ene, Cholesta-3,5-diene, Stigmasta-5,22-dien-3-ol, acetate, Sitosterol, Cholesta-3,5-dien-7-one, Stigmasta-3,5-dien-7-one        | $1.2 \pm 0.9$          | 70  | 1.1 | 1.10 | 0-4.3    |
| Tocopherols        |                                                                                                                                            |                        |     |     |      |          |
| Tocopherols        | γ-Tocopherol, α-Tocopherol                                                                                                                 | 0.3 ± 0.3       | 106 | 0.2 | 1.75 | 0-1.5    |
| Hopanoids          |                                                                                                                                            |                        |     |     |      |          |
| Hopanoids          | Trinosphopane, Norhopene, 22,29,30-trisnorhop-17(21)-ene, 22,29,30-trisnorhop-16(17)-ene, Norhopane, 25-norhopene                          | $1.3 \pm 0.4$          | 31  | 1.4 | 0.94 | 0.2-1.9  |
| (Poly)aromatics    |                                                                                                                                            |                        |     |     |      |          |
| Benzene            | Benzene                                                                                                                                    | 0.9 ± 0.4       | 43  | 0.8 | 1.14 | 0.4-2.5  |
| Benzaldehyde       | Benzaldehyde                                                                                                                               | 0.6 ± 0.3       | 41  | 0.6 | 1.08 | 0.3-1.5  |
| Acetylbenzene      | Acetyl-benzene                                                                                                                             | $\textbf{1.1}\pm0.4$   | 39  | 1.0 | 1.10 | 0.6-2.3  |
| Alkylbenzenes C3-9 | Ethyl-methyl-benzene, Benzene C7, Benzene C9,                                                                                              | $\pmb{1.9} \pm 0.5$    | 23  | 1.8 | 1.07 | 1.4-3.5  |
| Polyaromatics      | Styrene, Indene, 1,2-dihydro-naphthalene, 2,3-dihydro-inden-1-one, 1-methyl-napthalene, 2methyl-napthalene, Biphenyl, Fluorene, Anthracene | $\textbf{1.4}\pm0.4$   | 27  | 1.3 | 1.04 | 0.8-2.1  |
|                    |                                                                                                                                            |                        |     |     |      |          |

**Table 2.** Summary statistics for the molecular composition of sediment OM given as the relative abundances (expressed in %) of the 41 groups of Py organic compounds, which belong to 13 classes of OM that are indicated by the grey shading *(following part)*

Table 3. Whole-lake and clusters average for a selection of elemental geochemical parameters and of ratios indicative of OM source types and their degradation status

|                                                 |                                     | SPECIFIC FEA                           | TURES IN GEO             | CHEMISTRY                  |                            |                                      |                                         |                          |                          |  |
|-------------------------------------------------|-------------------------------------|----------------------------------------|--------------------------|----------------------------|----------------------------|--------------------------------------|-----------------------------------------|--------------------------|--------------------------|--|
|                                                 |                                     | Whole-lake a                | Near-shore
sites      | North/East
basins       | Shallower                  | South basin
Intermediate
depth | Deeper                                  | Shallow c                | entral areas             |  |
|                                                 |                                     |                                        | Cluster geo 4 | Cluster geo 1   | Cluster geo 6   | Cluster geo 2             | Cluster geo 5                | Clus                     | ter geo 3     |  |
|                                                 |                                     | $(n^{b}=42)$                           | (n=4)                    | (n=13)                     | (n=10)                     | (n=8)                                | (n=3)                                   | (n                       | =4)                      |  |
| Water de                                        | enth (m)                            | $9+7(78\%)^{c}$                        | 4 + 2             | 5+3                        | 8+3                        | 15 + 4                               | 24 + 1                           | 2                        | +1                       |  |
| Bulk densi                                      | ty (g cm -3 )            | $0.06 \pm 0.02 (33\%)$                 | 0.06 ± 0.03       | 0.07 ± 0.02         | 0.07 ± 0.02         | 0.05 ± 0.01                   | 0.026 ± 0.009                    | 0.10                     | $\pm 0.02$               |  |
| [bSi]                                           | (%)                                 | 13 + 6(46%)                            | 12 + 6                   | 13 + 3                     | 15 + 7                     | 7+3                                  | 4.2 + 0.3                               | 21                       | + 4                      |  |
|                                                 | (%)                                 | 38 + 10(26%)                           | 50 + 12           | 39 + 5                     | 34 + 7              | 37 + 4                               | 52 + 2                           | ${20\pm 8}$              |                          |  |
| [S] (ms                                         | o ko -1 )                | 11876 + 5920 (50%)                     | 17510 + 833       | 11683 + 3440        | 7550 + 1900                | 12896 + 3315                  | 26227 + 4833                            | 4879                     | +148                     |  |
| [ Br ] (m                                | σ kσ -1 )                | 149 + 35(23%)                          | 130 + 6                  | 153 + 36            | 145 + 35                   | 154 + 19                             | 204 + 26                                | 116                      | $\frac{1}{1} + 32$       |  |
| [Cu] (m                                         | -55 )
ng kg -1 )      | 34 + 13(38%)                           | 36 + 5                   | 28 + 6                     | 30 + 7                     | 42 + 6                               | 65 + 10                          | 24                       | +13                      |  |
| [Uu] (m
[Ni] (m                              | $r_{\rm s}$ $r_{\rm s}$ $r_{\rm s}$ | 19 + 5(25%)                            | 20 = 3
21 + 1         | 18 ± 4              | 17 + 2              | 12 = 0
21 + 4                     | $\frac{32}{27+1}$                       | 12                       | $\frac{1}{2}$ + 4        |  |
| [H] (II                                         | $a k a^{-1}$                        | 337 + 202(60%)                         | 407 + 141                | 251 + 47                   | 230 + 69                   | 427 + 94                             | 917 + 212                        | 203                      | + 87                     |  |
| [ Z n] (m                                | ·σ κσ·1)                            | $219 \pm 108 (49\%)$                   | 279 + 31          | $212 \pm 68$               | 139 + 42                   | 305 + 86                             | 417 + 33                                | 63                       | + 16                     |  |
| [21] (11
[Fe]                                | (%)                                 | 5 + 3(60%)                             | 3.1 + 2.1                | $2.7 \pm 1.7$              | $3.6 \pm 1.5$              | $9.1 \pm 2.4$                        | 4.3 + 2.2                               | 5.5 ± 1.7         |                          |  |
| [FC] (70)
Fo·Al                              |                                     | $1.5 \pm 0.8(53\%)$                    | $1.0 \pm 0.5$            | $1.0 \pm 0.6$              | $1.1 \pm 0.3$              | $2.5 \pm 0.9$                        | $1.3 \pm 0.6$                           | 1.9 ± 0.3         |                          |  |
| $[\Delta s] (mg kg^{-1})$                       |                                     | 35 + 20(57%)                           | 27 + 17                  | 26 + 16                    | $25 \pm 11$                | 64 + 11                              | 48 + 14                                 | 20                       | ) + 9                    |  |
| [P] (m                                          | σ kσ -1 )                | 1624 + 741 (46%)                       | 927 + 240                | 1065 + 295                 | 2088 + 730                 | 2074 + 275                           | 2766 + 869                              | 1224                     | $\frac{1}{1}$ + 216      |  |
| [ M n] (n                                | $r_{\rm s}$ $r_{\rm s}$ $r_{\rm s}$ | 729 + 1690(231%)                       | 162 + 53                 | $1000 \pm 290$
182 + 67 | 184 + 50                   | 305 + 93                             | 171 + 13                         | $5700 \pm 1597$          |                          |  |
| [iiii] (ii
Mn                                | ·Fe                                 | $0.02 \pm 0.03 (150\%)$                | $0.007 \pm 0.002$        | $0.008 \pm 0.003$          | $0.006 \pm 0.002$          | 0.004 + 0.001                        | $0.005 \pm 0.002$                       | $0.111 \pm 0.051$        |                          |  |
| [Co] (m                                         | μα kα -1 )               | 19 + 15(79%)                           | 15 + 8                   | 12 + 6                     | 13 + 5                     | $26 \pm 11$                          | 14 + 2                                  | 49 + 24                  |                          |  |
| [C0] (m
[Ph] (m                              | ις κς )
ος κσ -1 )    | 192 + 90(47%)                          | $19 \pm 6$               | $132 \pm 53$               | $15 \pm 5$
$115 \pm 42$ | $300 \pm 59$                         | $14 \pm 2$
$315 \pm 7$               | 187                      | $49 \pm 24$
182 + 96  |  |
|                                                 |                                     | SPECIFIC FEAT                          | TURES IN OM C            | OMPOSITION                 |                            | 000 = 07                             | 010 = 7                                 |                          |                          |  |
|                                                 |                                     |                                        | Near-shore               | North/Fast                 |                            | South basin                          |                                         | Shallow c                | entral areas             |  |
|                                                 |                                     | Whole-lake                             | sites                    | hasing                     | Shallowar/inta             | rmediate denth                       | Deener                                  | Shanow e                 | citit di di cus          |  |
|                                                 |                                     | WHOIC-IAKC                             | Cluster 5                | Cluster 1                  | Clust                      | or 3                                 | Cluster 2                               | Cluster 1                | Cluster 6                |  |
|                                                 |                                     | (n - 42)                               | (n-4)                    | (n-16)                     | Ciusi                      | (14)                                 | $Cluster_{OM} 2$                        | $Cluster_{OM} + (n-2)$   | (n-2)                    |  |
| Weden Jan                                       |                                     | (n-42)                                 | (n-4)                    | ( n -10)            | ( n -               | -14)                                 | (n-3)                                   | (n-3)                    | (n-2)                    |  |
| vvater dep                                      | ( 0 ()                       | $9 \pm 7 (70\%)$
$38 \pm 10 (26\%)$ | $4 \pm 2$                | $7 \pm 3$                  | 11                         | ± 3                                  | 24.1 $\pm$ 0.3 52 $\pm$ 2 | $3.2 \pm 0.9$            | $1.0 \pm 0.1$            |  |
| (C22, 25:0 + 2K, C22, 21): Liquing d | (70)
In Jaka Tempetrial plant OM | $30 \pm 10(2070)$                      | $50 \pm 12$              | 39 ± 4                     | 30                         | ± 5                                  | $34 \pm 2$                              | 24 ± 4                   | 14 ±0             |  |
| (C23-35:0+2K C23-31): Lightins                  | Alash Diant OM                      | $2 \pm 1 (30\%)$                       | $0.0 \pm 0.3$            | $5 \pm 1$                  | 1. / :              | ± 0.4                                | $1.0 \pm 0.0$                           | $3 \pm 1$                | $19 \pm 11$              |  |
| N-compounds : Carbonydrates                     | Algal:Plant OM                      | $0.37 \pm 0.09 (24\%)$                 | $0.32 \pm 0.08$          | $0.35 \pm 0.04$            | 0.39                       | ± 0.05                               | $0.0 \pm 0.1$                           | $0.29 \pm 0.02$          | $0.23 \pm 0.03$          |  |
| Chiorophylis : Plant lipids+lightins            | Algal: Plant OM                     | $0.18 \pm 0.09 (30\%)$                 | $0.10 \pm 0.03$          | 0.13 ± 0.00         | 0.24 :                     | ± 0.08                               | $0.31 \pm 0.07$                         | $0.10 \pm 0.03$          | $0.03 \pm 0.03$          |  |
| (alkyl)pyridines+Aromatic N                     | Algal OM (N-compounds) freshness    | 0.3 ± 0.1 (33%)                 | 0.39 ± 0.09       | $0.36 \pm 0.05$            | 0.22                       | ± 0.06                               | $0.42 \pm 0.06$                         | $\textbf{0.20} \pm 0.08$ | $\textbf{0.13} \pm 0.08$ |  |
| Phytadienes:pristenes c              | Algal OM (chlorophylls) freshness   | 0.4 ± 0.1 (25%)                 | 0.4 ± 0.1         | $\textbf{0.37} \pm 0.09$   | 0.40                       | ± 0.06                               | 0.56 ± 0.05                      | $\textbf{0.42} \pm 0.07$ | 0.5 ± 0.2         |  |
| Anhydrosugars:(alkyl)furans &
furanones      | Plant OM (carbohydrates) freshness  | 0.2 ± 0.2 (100%)                | 0.4 ± 0.2         | 0.3 ± 0.2           | 0.12                       | ± 0.11                               | $\textbf{0.14} \pm 0.04$                | $\textbf{0.08} \pm 0.01$ | $0.042 \pm 0.002$        |  |
| Guaiacyl-acid:Guaiacyl-aldehyde e    | Plant OM (lignin) freshness         | 0.07 ± 0.03 (43%)               | $0.13 \pm 0.02$          | 0.07 ± 0.03         | 0.05                       | ± 0.02                               | $\textbf{0.04} \pm 0.01$                | $0.04 \pm 0.03$          | $0.10 \pm 0.06$          |  |
| Guaiacyl -2C: Guaiacyl -1C e         | Plant OM (lignin) freshness         | 0.8 ± 0.3 (38%)                 | $1.23 \pm 0.07$          | $1.0 \pm 0.2$              | 0.5                        | ± 0.2                                | 0.6 ± 0.2                        | 0.5 ± 0.1         | $1.1 \pm 0.2$            |  |
| Svringvl-2C·Svringvl-1C e            | Plant OM (lignin) freshness         | $10 \pm 08(80\%)$                      | $2.4 \pm 0.3$            | $11 \pm 0.6$               | 0.5                        | +02                                  | $0.6 \pm 0.1$                           | 03 + 03                  | 14 + 08                  |  |

 Syringyl-2C:Syringyl-1C°
 Plant OM (lignin) freshness
 1.0  $\pm$  0.8 (80%)
 2.4  $\pm$  0.3
 1.1  $\pm$  0.6
 0.5  $\pm$  0.2
 0.6  $\pm$  0.1
 0.3  $\pm$  0.3
 1.4  $\pm$  0.8

 a whole-lake: averages of all analyzed sediment samples are calculated without the excluding the two outlier samples (sites M4, S15; cf. Sect. 2.53.1.1); b n: number of samples; c the data are presented as follow: average  $\pm$  standard deviation (relative standard deviation); d the compounds included in the ratios are given in detail in Table S1 in the supplementary information.

Light grey background denotes average values below whole-lake average (<10%); No background denotes values close to whole-lake average ( $\pm 10\%$ ); Dark grey background are values above whole-lake average (>10%).

**Figures**

---

## Editor Decision (ED1)

Dear Julie Tolu and co-authors,

First I would like to thank the two reviewers for their extensive review of your manuscript and of course you for your thorough reply. Without going into all the details of the reviews there are a few things I would like to emphasize here a bit more.

In reply to some of the comments of referee one you stated that a more extensive discussion on the Fe, Mn, and Al distributions in the lake falls outside the scope of this study. The scope of this study more or less defined as determining the spatial heterogeneity in the lake indicating that taking a single sample and scaling up to the entire surface area of the lake is not the way to go. I do agree that an extensive discussion is not warranted here, but you do suggest some mechanisms that might play a role and you specify these mechanisms. This makes the argument of the scope of the study invalid, I think, if something falls outside the scope of the work you could argue it should not be mentioned at all. It is not a very good excuse. Therefore I would like to ask you to either make the effort of writing a more extensive discussion on different possible mechanisms or you make it very clear that the mechanisms mentioned are just of few possibilities and that there might be others such as … (see review and reply).

The other thing I noticed is that the "whole-lake" concept creates some response from both reviewers in different ways. As for the title I agree with the suggested changes. Throughout the manuscript, just realize that you did not analyze the whole lake. Rather than just the typical one sediment sample you analyzed more samples from different sites to get an idea of the heterogeneity. This is very interesting, and provides a much better picture than analyzing just one sample, but you did not analyze the whole lake or even the all the lake sediments. One question I had is what about micro-heterogeneity did you analyze multiple "replicates" from the same sample to see how large the variation is within one sample? 200µg is a relatively small amount.

You suggest that you looked at the whole matrix rather than only specific compounds or classes. And although I agree that pyrolysis might be one of the better tools to do this, the method still has biases and issues like all other analytical methods. If it doesn't fit in your analytical window you don't see it. The discussion about the pyrolysis temperature makes that quite clear, different temperatures show (slightly) different results, so please be careful in how you phrase what your pyrolysis results reflect. That brings me to a comment from referee two about other studies on OM composition studies. Other groups have investigated complex organic matter from lake and other sediments through a whole range of different techniques including pyrolysis (with and without TMAH), lipid extractions, GC and LC mass spectrometry etc etc. I think your approach of analyzing every samples the same way works fine for this manuscript, but I think it is good to put your own work in a larger, historical perspective by shortly mentioning other studies. Pyrolysis has been around for quite some time and it is good that this would be reflected in your reference list.

As referee two mentions, be very careful with steroids and hopanoids, they come very different sources. Looks can be deceiving. Be very careful with assigning different classes and binning compounds together and using only or mainly the NIST database for compound identifications. Mistakes are easily made.

Based on the reviews and responses I think this manuscript can be published in Biogeosciences if the authors make the changes they indicated they would in their response with special attention to the items I mentioned above.

Best regards,

Marcel

---

## Author Response (AR2)

Dear Editor,

We have thoroughly checked the text of the whole manuscript and correct grammar/typing error as well as improve the clarity of sentences that were long. We would like to thank you for the few textual issues you pointed out; they have all been considered (as detailed below). We would be really happy to have our paper published in *Biogeosciences* because we believe our manuscript to be very well in the scope of this journal, and thus of interest for the readers of this journal.

Sincerely yours,

On the behalf of my co-authors, Julie Tolu

**Answer to the textual issues noted by the Editor:**

Line 59: Why not replace bacterial with micro-organisms? Probably it will need to be "derived from residues of plants, animals, fungi and micro-organisms", that way it also includes archaea and viruses and it separates the micro algae from the plants. => Yes, the term "micro-organisms" is more appropriated here.

Line 111: the space after 2009 and no space after the ; => Corrected

Line 133: space between pathway and ,? => Corrected

Line 160: define Spheroidal Carbonaceous Particle.

=> SCP has been replaced by « Spheroidal Carbonaceous Particle ». We remove the abbreviations because we are not using it thereafter in the manuscript.

Line 220-224: Why not something like "Pyrolysis at 450 °C is preferred to pyrolysis as 650 °C because it avoids complete degradation of some source specific biomarkers and enables determination of degradation status by analysis of Py products of polysaccharides and/or cellulose, syringol lignin oligomers, for instance (Tolu et al., 2015)"? => We have re-written the sentence following the Editor recommendation.

Line 279: and to explore => Corrected

Line 291: they are instead of there are? They are not significantly correlated... or there are no significant correlations... => Corrected

Line 308: is not confined to a specific mineral phase without the s (phases) => Yes, the "s" has been removed

Line 319: the SI => Corrected accordingly everywhere in the manuscript

Line 331: northern, eastern and central areas => Corrected

Line 352: I think it's McClymont with a capital C, same for McInerney (line 466). Check the Mc's.

=> All "Mc's" have been checked and corrected

Line 362: side-chain is or side-chains are, in the first case I will remain contains, for the latter it would be contain (363).

=> The sentence has been corrected.

Line 364: "pyran" compounds have 5 C and 1 O atoms, the heterocycle of dianhydrorhamnose ... consist of.. 6 C's and 1 O. The levosugars ... and levoglucosenone contains a carbonyl group.

=> We have corrected the sentence by adding "atoms" after "5 C and 1 O" but we did not change "the heterocycle of pyran compounds has" by "pyran compounds have", because it is only the heterocycle of pyran compounds that has 5 C and 1 O atoms and not the entirely structrue of the pyran compounds we identified.

Line 370: in the SI => Corrected

Line 381: micro-organism exoskeletons, no s after micro-organism. Now micro-organism is typically used for micro algae, bacteria and archaea (and a few others). I have the impression you mean zooplankton here? Perhaps it would better to state that to avoid confusion. Chitin from small/micro zooplankton exoskeletons...

=> The sentences has been corrected as followed: "Pyrolytic compounds containing an acetamide functional group previously shown to be a good indicator of the presence of chitin, a component of fundi cell walls and arthropod exoskeletons, in biological and geological samples (Gupta et al., 2007)"

Line 384: present also in the highest abundances in the three deepest sampling locations.

=> The proposed correction is not what we want to say. The sentence has, however, been modified as follows: "These three deepest sampling locations also present the highest abundances of pyrolytic compounds containing an acetamide functional group previously shown to be a good indicator of the presence of chitin, a component of fundi cell walls and arthropod exoskeletons, in biological and geological samples (Gupta et al., 2007), phytadienes (i.e., pyrolytic products of chlorophylls; Nguyen et al., 2003), short-chain alkan-2-ones (2K C13-17) and steroids."

Line 405: you can delete existing => Corrected

Line 412: defined rather than extracted? Extracted in a paper on OM might be confusing. => the term "extracted" that was used to refer to the number of selected PCA to describe PCA outputs has been replaced with "retained"

Line 421: A short summary sentence on the positive loadings is missing. For all the loadings you end with a short "summary". It seems that the positive loading is correlated with.... and... Something like line 432 and 433: Thus, negative ....

=> The sentence "Therefore, positive PC1OM scores represent samples rich in degraded OM" has been added as a summary for  $PC1_{OM}$  positive side, as done for the negative side of PC1 and the others PC

Line 445: long chain n-alkanes are usually related to plant wax lipids not plant cell wall lipids and besides by water they can also be transported by air/wind.

=> Yes, this was an error. Now we have decided to employ the generic term "lipids".

Line 452: hopanoids derived from prokaryotes, mainly bacteria. => Corrected as recommended.

Line 455: in lake algae and micro-organisms. I assume in lake micro algae, diatoms and such, and zooplankton. Micro-organisms also includes things like bacteria and archaea, a much broader term.

=> Here, we wanted to use the broad term. The PC4 positive side which is summarize in this sentence is associated to bacteria OM (see the sentence just above).

Line 466: Check the Mc... spelling

=> Mc's spelling has been checked and corrected for the whole manuscript.

Line 492: to avoid confusion "these shallower and intermediate water depth sites in the south basin...

=> Instead, the sentence has been rewritten for better clarity.

Line 512: cell wall? Plant wax lipids! => corrected

Line 555 accumulation at the bottom? Accumulation bottoms?

=> Our writing is correct, we are talking about different sediment bottom types of sediment focusing model, i.e. erosion bottom, transportation bottom and accumulation bottom. However, we have changed this term to be sure readers will understand properly.

Line 558: such as the surrounding lakes.... => Corrected

Line 567: is there no water column primary production that sinks to the bottom? At the deepest places this would integrate the largest water column and therefor the most "fresh" organic matter? Sinking particales!

=> Yes, this is also probably happening and we actually talked about it in the last (concluding paragraph) of the sub-section 3.3.1. In these specific lines, we want to mainly discuss the fact that even if there are substantial benthic algal production in shallower areas, as shown by bSi content and strongly supported by literature, the OM, including the algal derived OM, is strongly degraded.

Line 753: McInerney. Oh hey this paper is on leaf wax n-alkanes (not leaf cell wall n-alkanes).

=> Corrected (see comments above)

Line 848: McClymont ⇔ Corrected